# Deep Ensembles Work, But Are They Necessary?

**Taiga Abe**[*1]    **E. Kelly Buchanan**[*1]    **Geoff Pleiss**[1]    **Richard Zemel**[1]

**John P. Cunningham**[1]
[1]Columbia University
{ta2507,ekb2154,gmp2162,jpc2181}@columbia.edu
zemel@cs.columbia.edu

## Abstract

Ensembling neural networks is an effective way to increase accuracy, and can often match the performance of individual larger models. This observation poses a natural question: given the choice between a deep ensemble and a single neural network with similar accuracy, is one preferable over the other? Recent work suggests that deep ensembles may offer distinct benefits beyond predictive power: namely, uncertainty quantification and robustness to dataset shift. In this work, we demonstrate limitations to these purported benefits, and show that a single (but larger) neural network can replicate these qualities. First, we show that ensemble diversity, by any metric, does not meaningfully contribute to an ensemble's uncertainty quantification on out-of-distribution (OOD) data, but is instead highly correlated with the relative improvement of a single larger model. Second, we show that the OOD performance afforded by ensembles is strongly determined by their in-distribution (InD) performance, and—in this sense—is not indicative of any "effective robustness." While deep ensembles are a practical way to achieve improvements to predictive power, uncertainty quantification, and robustness, our results show that these improvements can be replicated by a (larger) single model.

## 1 Introduction

In many real-world settings, practitioners deploy ensembles of neural networks that combine the outputs of several individual models [e.g. 71, 44, 77]. Though training and evaluating multiple models is computationally expensive, a wide body of research demonstrates that ensembles achieve better performance (as measured by accuracy, negative log likelihood, or a variety of other metrics) than their constituent single models, provided that these models make diverse errors [16]. This benefit is well-established in the literature: theoretically proven for ensembles formed via boosting or bagging [67, 6], and demonstrated for *deep ensembles* that solely rely on the randomness of SGD coupled with non-convex loss surfaces [46, 18].

Of course, ensembling is not the only way to increase performance; one could also increase the depth or width of a single neural network. In many settings, a single large model performs similarly to an ensemble of (smaller) models with a similar number of parameters [48, 40, 74]. This observation poses a natural question: are there reasons to choose a deep ensemble over a single (larger) neural network with comparable performance?

Recent research suggests that deep ensembles may be preferable to single models in safety-critical applications and settings where data shifts significantly away from the training distribution. First,

---

[*]Equal contribution.

36th Conference on Neural Information Processing Systems (NeurIPS 2022).

Lakshminarayanan et al. [45] demonstrate that deep ensembles provide *well-calibrated estimates of uncertainty* on classification and regression tasks. Compared with other uncertainty quantification (UQ) methods, ensembles offer better (i.e. less overconfident) uncertainty estimates on out-of-distribution (OOD) or shifted data [62]. Second, recent work indicates that—beyond calibration— ensemble performance (as measured by accuracy, NLL, or other metrics) also tends to be *robust against dataset shift*, again often outperforming other methods in these regimes [27].

Intuitions in recent papers [e.g. 46, 18] attribute these UQ/robustness benefits to the fact that ensembles produce multiple diverse predictions, rather than a single point prediction. If diversity does in fact explain UQ/robustness improvements, this would suggest that deep ensembles indeed offer benefits that cannot be obtained by (standard) single neural networks. In this paper, we rigorously test hypotheses that formalize this intuition. Surprisingly, after controlling for factors related to the performance of an ensemble's component models, we find no evidence that having a diverse set of predictions is responsible for these purported benefits. Put differently, we find that these UQ/robustness benefits are not unique to deep ensembles, *as they can be replicated through the use of (larger) single models*. We confirm these results for a wide variety of model architectures, as well as for *heterogeneous deep ensembles* that combine multiple different neural network architectures and *implicit deep ensembles* like MC Dropout [21], BatchEnsemble [75], and MIMO [30] (Appx. H.4).

**Hypothesis: ensemble diversity is responsible for improved UQ.** Two components contribute to ensemble uncertainty estimates: the uncertainties expressed by individual ensemble members, and diversity among ensemble member predictions. Recent work suggests that the diversity component is primarily responsible for better calibrated OOD uncertainty estimates, as ensemble members should agree less (i.e. offer more diverse predictions) as data shift away from the training distribution [45, 18, 27]. In contrast, we find that—after conditioning on the uncertainty of individual ensemble members—*the level of ensemble disagreement does not statistically differ between InD and OOD data* (Fig. 1), and thus ensemble diversity is not directly responsible for larger OOD uncertainty estimates. Furthermore, ensemble diversity—on a per-datapoint basis—is correlated with the expected improvement we obtain by increasing model capacity (Fig. 2), implying that ensemble diversity does not capture a quantity inaccessible to a single (larger) model.

**Hypothesis: ensemble diversity is responsible for improved robustness.** Independent work demonstrates a deterministic relationship between a (single) neural network's 0-1 accuracy on InD and OOD datasets [72, 52], whereby the OOD performance of a model can be predicted from its InD performance. It is therefore natural to ask whether having multiple diverse predictions contributes to additional OOD robustness (as suggested by [18, 27]), beyond what is expected given performance improvements on InD data. Our results demonstrate that deep ensembles are not "effectively robust" relative to single models—i.e. their OOD performance (as measured by accuracy, NLL, Brier score, and calibration error) follows the same deterministic relationship to InD performance as single models (Fig. 4). Therefore, ensemble diversity does not yield additional robustness over what standard single networks achieve.

**Implications.** Overall, this paper does not disagree with prior claims about the benefits of deep ensembles relative to an ensemble's component models. Indeed, in our experiments we confirm that ensembling is a convenient mechanism to improve predictive performance, UQ, and robustness relative to this baseline. At the same time, our results also indicate that—after controlling for individual model uncertainty and InD performance— ensembles do not obtain UQ/robustness benefits beyond what can already be obtained from the properties of an appropriately chosen single model.

## 2   Related work

Ensembling is an established technique to improve generalization [e.g. 67, 63, 17, 60], where the predictions of multiple models are aggregated to reach a consensus. It is well established that diversity amongst ensemble members is necessary to improve performance [16]. This diversity can be achieved through many means. Randomization approaches introduce diversity by training each model on a random subset of data [6] or a random subset of features [7]. Alternatively, boosting approaches [19, 20] achieve diversity by manipulating the weighting of training data. Other methods include using a diverse set of model classes [e.g. 10] or joint training objectives [e.g. 55].

**Ensembles of neural networks.** Historically, neural network ensembles have relied on a variety of mechanisms to introduce diversity [e.g. 28, 63, 54, 79]. Recently, diversity is often obtained by

training multiple copies of the same neural network architecture with different intializations and minibatch orderings, as the inherent randomness of SGD has been shown to introduce a sufficient amount of diversity in these (non-convex) models [46, 24, 18]. Importantly, this approach can exploit parallel computation [45], because none of the ensemble members depend on one another.

**Deep ensembles for predictive uncertainty.** It has been suggested that ensembles of neural networks not only improve accuracy but also estimates of predictive uncertainty [45]. Some research aims to connect ensembles and Bayesian neural networks, suggesting that these improved uncertainty estimates are the result of performing approximate Bayesian model averaging [21, 76]. Although prior work has described shortcomings in the uncertainty estimates derived from deep ensembles [e.g. 47, 11, 32, 61], they remain a gold standard in high risk and safety critical settings [e.g. 62, 27, 73].

**Deep ensembles and robustness.** Robustness is the ability to maintain good accuracy and calibration under conditions of distributional shift. Deep ensembles outperform other approaches in maintaining both accuracy and calibration on OOD data [62, 27], although their limitations have also been demonstrated [43, 64]. This robustness is attributed to the diversity between ensemble members [18].

**Other related work.** Recent work investigates whether it is possible to achieve the benefits of an ensemble with reduced computation during training and/or test time [36, 49, 75, 30]. Additionally many works have proposed numerous diversity metrics for ensembles similar to those we examine here [e.g. 42, 51, 3].

# 3   Setup

Consider multiclass classification: inputs $\boldsymbol{x} \in \mathbb{R}^D$ with targets $y \in [1, \ldots, C]$, where $D$ is the number of features and $C$ is the number of classes. We assume that we have access to $M$ distinct neural networks $\boldsymbol{f}_1, \ldots, \boldsymbol{f}_M$, where each model $\boldsymbol{f}_i : \mathbb{R}^D \to \Delta^C$ maps an input to the $C$-class probability simplex. We will primarily focus on the common case of **homogeneous ensembles**, where $\boldsymbol{f}_1, \ldots, \boldsymbol{f}_M$ represent the same neural network architecture and training procedure, relying on the inherent randomness of initialization and SGD to produce diverse models (see Sec. 2 for a broad discussion). However, in Sec. 5.3 we will also consider **heterogeneous ensembles** where $\boldsymbol{f}_1, \ldots, \boldsymbol{f}_M$ represent different architectures or training procedures, and **implicit ensembles**, where $\boldsymbol{f}_1, \ldots, \boldsymbol{f}_M$ are approximated by changes to a single model [21, 30, 75]. Throughout the paper, we will also represent these member networks as a discrete distribution of models: $p(\boldsymbol{f}) = \text{Unif.}[\boldsymbol{f}_1, \ldots, \boldsymbol{f}_M]$. The ensemble prediction $\bar{\boldsymbol{f}}(\boldsymbol{x})$ is given by the arithmetic mean of the ensemble member *probabilities*:[1]

$$\bar{\boldsymbol{f}}(\boldsymbol{x}) = \mathbb{E}_{p(\boldsymbol{f})}[\boldsymbol{f}(\boldsymbol{x})] = \tfrac{1}{M} \sum_{i=1}^{M} \boldsymbol{f}_i(\boldsymbol{x}) \tag{1}$$

**Metrics for ensemble diversity.** Two metrics of ensemble diversity are (1) variance [e.g. 39], and (2) Jensen-Shannon divergence [e.g. 45, 18]. Mathematically, they are (respectively) defined as:

$$\operatorname*{Var}_{p(\boldsymbol{f})}[\boldsymbol{f}(\boldsymbol{x})] = \sum_{i=1}^{C} \operatorname*{Var}_{p(\boldsymbol{f})}\left[f^{(i)}(\boldsymbol{x})\right], \quad \operatorname*{JSD}_{p(\boldsymbol{f})}[y \,|\, \boldsymbol{f}(\boldsymbol{x})] = \text{H}\left[y \,|\, \bar{\boldsymbol{f}}(\boldsymbol{x})\right] - \mathbb{E}_{p(\boldsymbol{f})}\left[\text{H}\left[y \,|\, \boldsymbol{f}(\boldsymbol{x})\right]\right] \tag{2}$$

where $f^{(i)}$ refers to the probability assigned by a model to the $i$-th output class, and H is the entropy. Both metrics are always positive and minimized when the predictions from ensemble members are the same, i.e. not diverse.

**Models and training datasets.** We reuse and train a variety of neural networks on two benchmark image classification datasets: **CIFAR10** [41] and **ImageNet** [14]. In particular, we include the 137 CIFAR10 models trained by Miller et al. [52], corresponding to 32 different architectures each trained for 2-5 seeds; as well as the "standard" 78 ImageNet models curated by Taori et al. [72], each corresponding to a different architecture trained for 1 seed. To form homogeneous ensembles, we additionally train 10 network architectures on CIFAR10 and three on ImageNet. We train 5 independent instances of each model architecture, where each instance differs only in terms of initialization and minibatch ordering. We form homogeneous deep ensembles by combining 4 out of the 5 random seeds. From this process, we can consider 5 single model replicas and 5 ensemble replicas for each model architecture. Unless otherwise stated, ensembles are formed following Eq. (1).

---

[1]While it is also possible to average the logits (log probabilities) of each model, we note that probability averaging is far more common in the literature [e.g. 45].

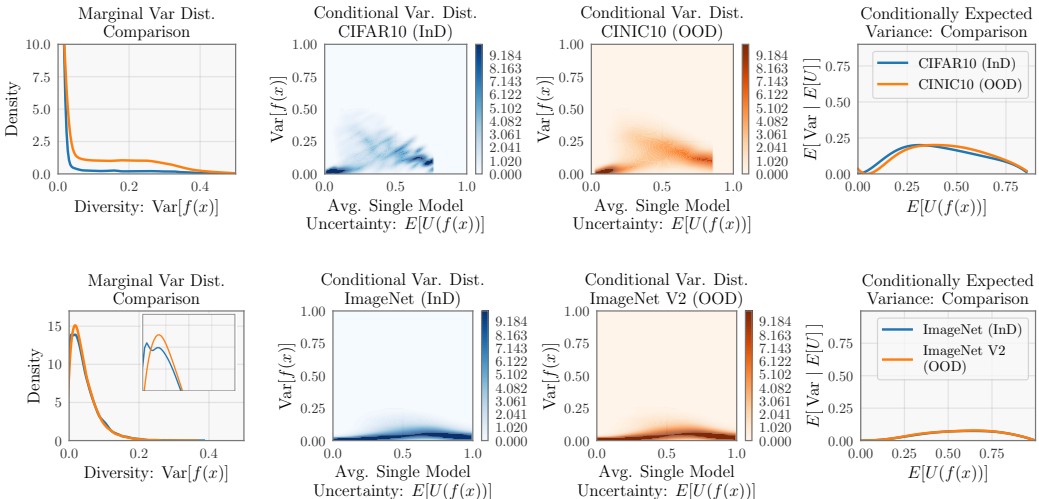

Figure 1: **Ensemble diversity does not yield better OOD uncertainty quantification, after controlling for average single model uncertainty.** Panels compare ensemble variance ($\mathrm{Var}[\boldsymbol{f}(\boldsymbol{x})]$) on InD (blue) vs. OOD (orange) data. The top row represents the variance for ensembles composed of 5 WideResNet 28-10 [78] networks evaluated on CIFAR10 and CINIC10, and the bottom row represents the variance for ensembles of 5 AlexNets, evaluated on ImageNet and ImageNetV2. The left column shows that, consistent with previous results, deep ensembles express higher variance predictions on OOD vs. InD data. The middle columns show $p(\mathrm{Var} \mid \mathbb{E}[U])$ (arguments suppressed for clarity) for InD (second column) and OOD data (third column). Suprisingly, we find that these conditional distributions are extremely similar. In the right columns, we further show the similarity of these conditional distributions (InD and OOD) using the conditional expectation $\mathbb{E}[\mathrm{Var} \mid \mathbb{E}[U]]$, estimated with kernel ridge regression. For experimental details, see Appx. F.

**OOD datasets.** A majority of our analysis compares deep ensembles on InD versus OOD test data. To that end, we consider three different catagories of OOD datasets as suggested by [52]: *Shifted reproduction datasets.* This category includes the **CIFAR10.1** and **ImageNetV2** datasets [66], both of which were collected and labeled following the same curation processes of the original CIFAR10 and ImageNet datasets, respectively. Neural networks (trained on the original datasets) tend to achieve worse performance on these new test sets. *Alternative benchmark datasets.* The **CINIC10** dataset [12] shares the same classes as CIFAR10 but uses images drawn and downsampled from the ImageNet dataset. Because ImageNet and CIFAR10 images were collected using different curation procedures, models trained on CIFAR10 tend to achieve worse performance on CINIC10. *Synthetically corrupted datasets.* The **CIFAR10C** and **ImageNetC** datasets [34], apply synthetic perturbations to CIFAR10 and ImageNet images (e.g. Gaussian blur, fog effects, etc.). Due to their synthetic nature, these datasets offer shifts of various intensity (e.g. mild blur versus heavy blur). We relegate most of our analysis of these datasets to the Appendix.

## 4 Hypothesis: ensemble diversity is responsible for improved UQ

The ability of deep ensembles to produce higher estimates of uncertainty on OOD data has been attributed to ensemble diversity [45, 18, 76]. In particular, ensemble diversity is hypothesized to increase on OOD data, where one would expect that OOD predictions from individual ensemble members are less constrained by their shared training data [45]. This hypothesis is attractive because it suggests that deep ensembles offer an additional mechanism for uncertainty quantification beyond what is afforded by any single model. In this section, we test this hypothesis by quantifying the contribution of ensemble diversity to a deep ensemble's total predictive uncertainty on both InD and OOD data.

## 4.1 Metrics for ensemble diversity

Common metrics for ensemble diversity provide interpretable decompositions of uncertainty: *ensemble uncertainty = ensemble diversity + average single model uncertainty*. For example, if we use variance (Eq. 2) as a metric for ensemble diversity [39, 27], then we show ensemble uncertainty can be decomposed as:

$$\overbrace{U\left(\bar{\boldsymbol{f}}(\boldsymbol{x})\right)}^{\text{ens. uncert.}} = \overbrace{\underset{p(\boldsymbol{f})}{\text{Var}}\left[\boldsymbol{f}(\boldsymbol{x})\right]}^{\text{ens. diversity}} + \overbrace{\underset{p(\boldsymbol{f})}{\mathbb{E}}\left[U\left(\boldsymbol{f}(\boldsymbol{x})\right)\right]}^{\text{avg. single model uncert.}} . \tag{3}$$

where $\boldsymbol{f}(\boldsymbol{x}) \in \Delta^C$ is a probabilistic prediction, and $U(\boldsymbol{f}(\boldsymbol{x}))$ is a quadratic notion of uncertainty:

$$U\left(\boldsymbol{f}(\boldsymbol{x})\right) \triangleq 1 - \sum_{i=1}^{C}\left[p(y = i \mid \boldsymbol{f}(\boldsymbol{x}))\right]^2.$$

See derivation in Appx. C. Intuitively, $U$ will be small when most probability is placed on a single class, and will be large when probability is distributed amongst classes. See Appx. C for analogous results with Jensen Shannon divergence as the diversity metric (Eq. 2). Based on our hypothesis, ensemble diversity (Var in Eq. 3) should increase on OOD data *independently* of average single model uncertainty ($\mathbb{E}[U]$). In other words, given *any* level of $\mathbb{E}[U]$, we would expect more ensemble diversity for OOD data than InD data.

## 4.2 Experiment: InD vs OOD ensemble diversity

We test 10 different ensembles of size $M = 5$ trained on CIFAR10, and three ensembles trained on ImageNet. We evaluate these ensembles on their respective InD (CIFAR10, Imagenet) and OOD (CIFAR10.1, CINIC10, CIFAR10C, ImageNet V2, ImageNetC) test sets. In Fig. 1, we analyze the variance of two of these deep ensembles, evaluated on CIFAR10 vs CINIC10 (top row) and ImageNet vs ImageNetV2 (bottom row), see Appx. F for a complete set of results. The left panel of Fig. 1 shows the distribution $p(\text{Var})$ for InD and OOD data. Ensembles tend to express higher variance on OOD data than InD data; a finding consistent with previous work [45, 18]. However, we emphasize this result is not sufficient to directly attribute UQ improvements to ensemble diversity.

**Controlling for single model uncertainty.** A different picture emerges when we control for single model uncertainty. Fig. 1 (middle) shows histograms of $p(\text{Var} \mid \mathbb{E}[U])$ i.e. the ensemble variance *conditioned on* average single model uncertainty as given by Eq. (3). Surprisingly, we see that the OOD and InD conditional distributions are very similar. We further study this similarity in Fig. 1 (right), which plots expected ensemble variance conditioned on average single model uncertainty: $\mathbb{E}[\text{Var} \mid \mathbb{E}[U]]$. Far from what our hypothesis would suggest (i.e. higher OOD diversity across all levels of average single model uncertainty) we observe that the conditional expectation of ensemble diversity on InD vs OOD data is nearly identical. In Appx. F (Fig. 8-Fig. 12), we offer statistical validation of these observations, and further demonstrate that this phenomenon holds across various architectures, InD, and OOD datasets. In all cases, the difference between the InD and OOD expected variance is only a few percentage points, and/or not statistically significant.

**Understanding the relationship between ensemble diversity and average single model uncertainty.** By controlling for average single model uncertainty, we see that ensemble diversity does not differ significantly for InD versus OOD data. In turn, these results imply that the InD/OOD difference we see in Fig. 1 (left) must be due entirely to a change in the distribution of *average single model uncertainty*, $p(\mathbb{E}[U])$. From these results, we can conclude that surprisingly, the UQ benefits of ensemble diversity are dictated by the corresponding average single model uncertainty. In Appx. F.1 we plot the differences in $p(\mathbb{E}[U])$ that drive the changes in ensemble diversity observed in Fig. 1 (left).

## 4.3 What does ensemble diversity actually measure?

Our analysis above shows that ensemble diversity is not directly responsible for the improved OOD uncertainty estimates offered by ensembles. To begin to understand why this might be the case, it is useful to consider the link between ensemble diversity and performance. It has long been established that diversity amongst ensemble members is a necessary and sufficient condition for the superior performance of ensembles [e.g. 16]. To demonstrate this, consider any strictly convex loss function,

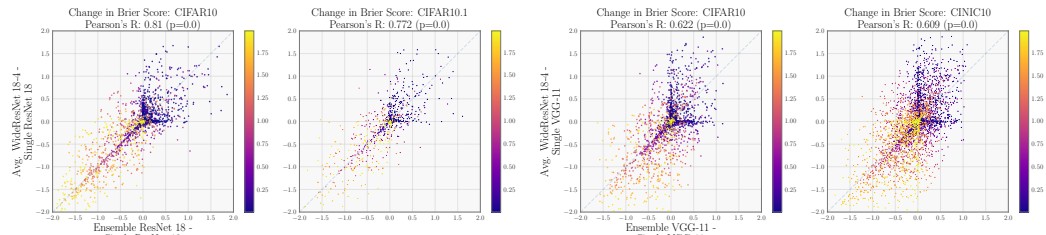

Figure 2: **Ensemble diversity is meaningfully correlated with the expected improvements from increasing model capacity**. (Left) Panels illustrate the per-datapoint gains in Brier score over a single ResNet 18 model by either forming a deep ensemble of ResNet 18 models (x-axis), or by increasing single model capacity, here with a WideResNet 18-4 (y-axis). The ResNet-18 ensemble and WideResNet 18-4 achieve nearly identical performance and strongly correlated improvements on both CIFAR10 and CIFAR10.1. Colors indicate the Brier score achieved by the single ResNet 18 model on each datapoint. (Right) We repeat the experiment for CIFAR10/CINIC10, showing the gains in Brier score over a VGG11 model, using either an ensemble of VGG11, or a WideResNet 18-4 model. Improvements are indistinguishable from relevant controls, and corresponding model accuracies are well matched, as shown in Appx. I.

such as negative log likelihood (NLL) or the multiclass Brier score (B) [8]:

$$\text{NLL}(\boldsymbol{f}(\boldsymbol{x}), y) \triangleq -\log\left(f^{(y)}(\boldsymbol{x})\right), \quad \text{B}(\boldsymbol{f}(\boldsymbol{x}), y) \triangleq \|\boldsymbol{f}(\boldsymbol{x}) - \mathbf{1}_y\|_2^2. \tag{4}$$

(Here, $\mathbf{1}_y$ represents a one-hot encoding of $y$.) Recall that the ensemble prediction $\bar{\boldsymbol{f}}(\boldsymbol{x})$ is the average of all model predictions (i.e. $\mathbb{E}_{p(\boldsymbol{f})}[\boldsymbol{f}(\boldsymbol{x})]$). By Jensen's inequality:

$$\text{NLL}(\bar{\boldsymbol{f}}(\boldsymbol{x}), y) \leq \mathop{\mathbb{E}}_{p(\boldsymbol{f})}\left[\text{NLL}(\boldsymbol{f}(\boldsymbol{x}), y)\right], \quad \text{B}(\bar{\boldsymbol{f}}(\boldsymbol{x}), y) \leq \mathop{\mathbb{E}}_{p(\boldsymbol{f})}\left[\text{B}(\boldsymbol{f}(\boldsymbol{x}), y)\right] \tag{5}$$

In other words, the performance of the ensemble (as measured by NLL or Brier score) must be better than the average performance of ensemble members. Because both NLL and Brier score are strictly convex, the Jensen gap in Eq. (5) will grow as $p(\boldsymbol{f})$ becomes less constant, or more "diverse." In particular, the Jensen gap for Brier score is exactly equal to the ensemble variance (Eq. 2):

$$\text{B}(\bar{\boldsymbol{f}}(\boldsymbol{x}), y) - \mathop{\mathbb{E}}_{p(\boldsymbol{f})}\left[\text{B}(\boldsymbol{f}(\boldsymbol{x}), y)\right] = \mathop{\text{Var}}_{p(\boldsymbol{f})}[\boldsymbol{f}(\boldsymbol{x})]. \tag{6}$$

(Similar results are well known in the regression context—[e.g. 42, 51]—see Appx. D for a short derivation). In other words, $\text{Var}_{p(\boldsymbol{f})}[\boldsymbol{f}(\boldsymbol{x})]$ measures the expected predictive improvement we obtain through ensembling. We can use these results to investigate our UQ findings. Hypothetically, if $\text{Var}_{p(\boldsymbol{f})}[\boldsymbol{f}(\boldsymbol{x})]$ were also responsible for improving UQ, this would imply that the performance gains from ensembling are somehow fundamentally different than the performance gains from increasing a single model's capacity, as the latter can hurt uncertainty estimates [26]. However, in the next section we demonstrate that these two methods of increasing performance are in fact correlated.

### 4.4 Ensembling versus increasing model capacity

In Fig. 2, we compare the expected per-datapoint performance improvement gained through ensembling (x-axis) to the performance improvement gained through increasing model capacity (y-axis). Specifically, we compare an ensemble of 4 CIFAR10 models (ResNet18) with a single large model (WideResNet-18-4). The ensemble and the large single model achieve comparable Brier Score: $0.084 \pm 0.002$ on the InD test dataset and $0.210 \pm 0.002$ on the CIFAR10.1 OOD dataset. In Fig. 2 (left), we plot the Brier score of the ensemble versus the large model on a per-datapoint level, depicting the *improvement correlation* across the dataset.

Surprisingly, we find that increasing model capacity and ensembling yield very similar performance improvements *on most datapoints*. The ensemble improvements and large model improvements have a Pearson's correlation of 0.81 on the InD test set. Importantly, we see that this correlation is preserved even on OOD data (Pearson's correlation: 0.76). We replicate this result for a different

ensemble/larger model pair (VGG-11 ensemble versus WideResNet-18-4) that again have nearly identical InD and OOD performance: $0.093 \pm 0.004$ CINIC10 InD Brier Score; $0.48 \pm 0.02$ CINIC10 OOD Brier Score (Fig. 2, right). We compare each improvement correlation in Fig. 2 to relevant controls, and ensure comparable accuracies (Appx. I). In all cases we find that improvements are as similar as we might expect if comparing two performance matched ensembles, or two single models. This result is unexpected, because the ensemble and the large model represent two distinct architectures (ResNet versus WideResNet) and two different modes of training (independent training of separate models versus training one large model). Recalling the relationship between ensemble diversity and relative performance gains, these results suggest that *ensemble diversity estimates the improvement we should expect by increasing model capacity.* We conclude that, with regards to UQ and performance improvements, ensemble diversity offers no significant benefit over what can be obtained with single models.

### 4.5 Implications for uncertainty estimation

**Epistemic vs. aleatoric uncertainty.** Uncertainty is often categorized as coming from one of two components [e.g. 38]. The *epistemic* component is said to capture uncertainty due to a limited number of observations, or uncertainty that the model accurately and uniquely captures the ground truth labeling process. Apparently, it can be reduced by collecting more data. In contrast, the *aleatoric* component is described as capturing the inherent ambiguity in the data (e.g. a blurry image) and is considered to be irreducible noise. In decision making applications such as active learning [68, 22] or model-based reinforcement learning [44, 77], this uncertainty decomposition is employed to identify informative datapoints for our model to sample next [15]. Previous work has interpreted ensemble diversity as in Eq. (2) as epistemic uncertainty [50, 27, 77], with average single model uncertainty in Eqs. (3) and (7) identified as aleatoric uncertainty correspondingly [70]. Our results in Fig. 1 demonstrate that there is a limitation to this interpretation, as we would expect more ensemble variance (the proxy for epistemic uncertainty) for OOD data than for InD data, independent of single model uncertainty (the proxy for aleatoric uncertainty). We therefore suggest caution when using ensembles to differentiate sources of uncertainty in downstream applications.

**Bayesian perspective.** Bayesian model averaging, or BMA integrates predictions against a posterior distribution over models. Given training data $\mathcal{D}$, BMA forms the prediction $p(y \mid \boldsymbol{x}, \mathcal{D}) = \int \boldsymbol{f}(\boldsymbol{x}) \, p(\boldsymbol{f} \mid \mathcal{D}) \, \mathrm{d}\boldsymbol{f}$. The advantage of BMA is the ability to consider all possible predictions given a prior and conditioned on training data, thereby mitigating the risk in estimating the "true" model from limited data. A recent line of work argues that modern deep ensembles (unlike classic ensembles—see Minka [53]) can be viewed as approximate BMA [35, 76], although we also note that concurrent work emphasizes differences between deep ensembles and Bayesian inference in the infinite width limit [31]. Our results in Fig. 1 identify a limitation of ensembles as approximate Bayesian inference. The posterior predictive distribution should express higher variance for OOD data than InD data, which is not the case for the deep ensemble predictive distribution. In Appx. E, we demonstrate that exact Bayesian inference does yield higher OOD posterior variance, even after conditioning on observational noise. We emphasize that our results neither agree nor disagree with the BMA interpretation of ensembling. Rather they suggest that ensemble members should not be interpreted as true posterior samples, and that (as with many approximate Bayesian methods) the ensemble approximation to BMA is biased.

## 5 Hypothesis: ensemble diversity is responsible for improved robustness

Beyond uncertainty quantification, ensembles have been shown to often achieve better predictive performance than single networks (as measured by 0-1 accuracy, NLL, or Brier score) on OOD or shifted datasets [45, 62, 27]. In this section, we test the hypothesis that ensemble diversity improves robustness over what single neural networks can offer.

### 5.1 Effective robustness

We use the concept of "effective robustness" as introduced by Taori et al. [72]. These authors note that there is often a deterministic relationship between a neural network's accuracy on InD data and its accuracy on an OOD dataset (green line in Fig. 3). In other words, any improvements in OOD performance can be entirely explained by improvements in OOD performance. A model is

considered to be *effectively robust* only if it achieves better OOD accuracy than what is predicted by its InD accuracy. In general, there are very few neural networks or training procedures that exhibit effective robustness against any OOD dataset [72, 52]. To measure the role that ensemble diversity plays in robustness, we quantify to what extent deep ensemble OOD performance can be explained by InD performance (as measured by the deterministic relationship derived from single models). If the performance of deep ensembles follows the same deterministic relationship, then deep ensembles are not effectively robust (i.e. multiple diverse predictors offer no additional robustness over what a single neural network provides).

## 5.2 Experiment: measuring effective robustness of deep ensembles across metrics

**Ensembles are not effectively robust with respect to 0-1 accuracy.** Following Miller et al. [52], we measure the InD and OOD error for all the models described in Sec. 3. The top left of Fig. 4 compares the error of models on CIFAR10 (InD) versus CINIC10 (OOD), and the bottom left plot compares the error of models on ImageNet (InD) versus ImageNetV2 (OOD). From these plots, we observe several trends. In agreement with Taori et al. [72] and Miller et al. [52], we observe that single models (green dots) follow a colinear relationship for InD versus OOD accuracy. Additionally, we find that *ensembles (orange dots) do not deviate from this colinear InD/OOD relationship*. In Appx. H.1, we evaluate the quality of these linear

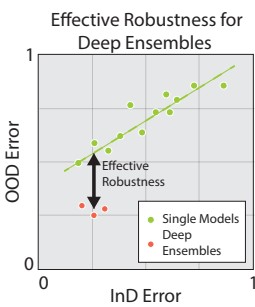

Figure 3: **Cartoon of effectively robust deep ensembles (what we want).**

trends. In particular, we fit separate linear trend lines for individual models and deep ensembles. All trend lines achieve correlations of $R > 0.84$, and their coefficients only differ by $1\%$ at most. This suggests that, after controlling for InD accuracy, the OOD accuracy of ensembles is nearly identical to that expected of single models. (See Appx. H for CIFAR10.1/CIFAR10C/ImageNetC results.)

**Ensembles are not effectively robust with respect to NLL or Brier score.** Although deep ensembles are not effectively robust in terms of predictive accuracy, many of their robustness benefits have been reported in terms of probabilistic metrics, such as NLL or Brier score [62]. We therefore extend our investigation of deep ensemble effective robustness to these metrics. Fig. 4 (middle left) plots the InD NLL and OOD NLL of various ensembles and single models. To the best of our knowledge, this is the first time that the effective robustness experiments of Taori et al. [72] and Miller et al. [52] have been extended to metrics other than 0-1 accuracy. We observe that the relationship between InD NLL and OOD NLL is not as linear as the accuracy trend. Nevertheless, we observe no discernible difference between the performance of single networks and ensembles (see Appx. H.1 for a quantitative analysis). We observe a similiar phenomenon when we plot InD versus OOD Brier score (Fig. 4, middle right)—ensembles and single models obtain similar OOD Brier score, after controlling for InD Brier score. Our key conclusion is that deep ensembles fail to demonstrate effective robustness when evaluated on probabilistic performance metrics, just as they do with 0-1 accuracy. (See Appx. H for CIFAR10.1/CIFAR10C/ImageNetC results.)

**Ensembles do not offer effectively robust calibration.** We also compare InD and OOD calibration for various single models and ensembles. We consider various metrics for measuring and comparing calibration used throughout the literature. Expected Calibration Error (ECE) [58] is a standard metric for measuring calibration of neural networks. As we show in Appx. H.3, there is little correlation between a single model's InD ECE and OOD ECE, which precludes any discussion of "effective robustness" using this metric. Conversely, Fig. 4 depicts a strong correlation between a model's InD/OOD square root of the Expected *Squared* Calibration Error (rESCE) [13, 56], which appears in a common decomposition of the Brier score [9]. We therefore expect that any InD/OOD trend for the rESCE should be qualitatively similar to the InD/OOD trends observed for Brier score. In Fig. 4 (top right), we observe a linear trend relating the CIFAR10 (InD) and CINIC10 (OOD) rESCE of single models. The rESCE of the ImageNet models, Fig. 4 (bottom right), follows a bimodal trend, where—depending on the model architecture—InD rESCE is correlated with either low or high OOD calibration. Nevertheless, for both datasets we find that ensembles do not achieve better OOD calibration that single models with similar InD calibration. (See Appx. H for CIFAR10.1/CIFAR10C/ImageNetC results.)

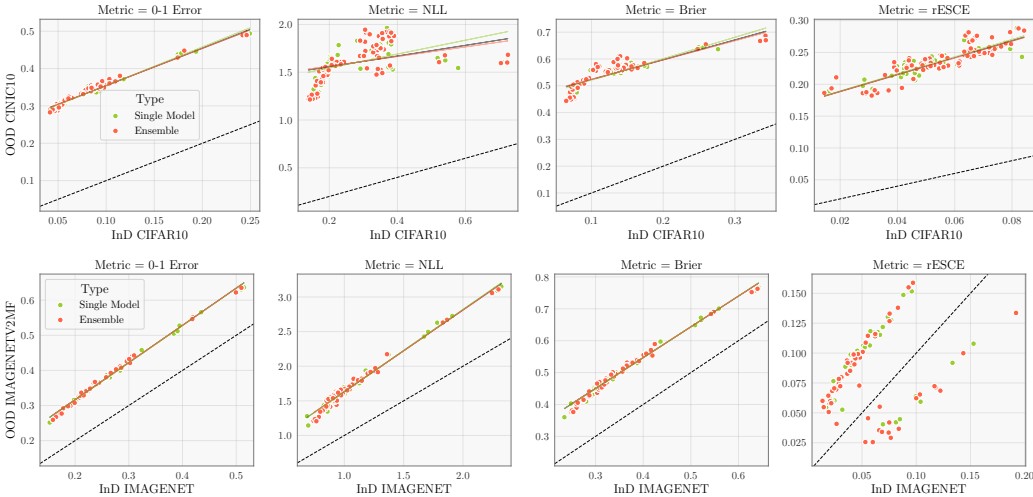

Figure 4: **Deep ensembles are not "effectively robust" across a variety of performance metrics**. Panels illustrate InD vs OOD performance metrics, from left to right: 0-1 Error, NLL, Brier Score, and rESCE. The model types considered are single models and ensembles. Linear trend lines are shown in solid lines, and black dotted lines indicate perfect robustness. We find that, conditioned on InD performance, ensembles offer no better OOD performance than single models. See Appx. H for additional corruptions.

## 5.3 Heterogeneous and implicit ensembles

From the previous results, it is clear that—by many metrics—ensembling multiple copies of the same model architecture confers no additional robustness over single models. A natural question is whether we could achieve more robustness by ensembling different model architectures together. To test this hypothesis, we repeat the same robustness experiments with *heterogeneous ensembles*: ensembles that combine multiple architectures, and *implicit ensembles*: single models that approximate deep ensembles, usually through parameter sampling [21]. To construct heterogeneous ensembles, we divide the 137 CIFAR10 models and 78 ImageNet models from Sec. 3 based on their InD accuracy. Ensembles are then formed by randomly selecting 4 models from each bin. This procedure ensures that all ensemble members will have similar accuracy, even though the ensemble members may represent different architectures and training regimens. Despite their additional diversity, these heterogeneous ensembles do not provide effective robustness, as shown in Appx. H.4. Finally, we investigate if these results also follow for three implicit ensembling mechanisms: Monte Carlo Dropout [21], multiple-input-multiple-output (MIMO) [30], and Batch Ensembles [75]. We find that implicit ensembles are also not effectively robust, as depicted in Appx. H.4.

## 5.4 Implications.

As discussed in Sec. 4.3, ensemble diversity is responsible for improved NLL and Brier score relative to constituent models. In this sense, ensemble diversity is responsible for improved OOD performance. However, these OOD improvements exactly follow the deterministic trends predicted by (standard) single models, and thus ensembling multiple diverse predictors does not yield any "effective robustness" over what could be achieved by a better performing single model. Unlike prior research [e.g. 62, 27], these results suggest that ensembles are a tool of convenience for obtaining better OOD performance, but not qualitatively different from single models in this respect.

## 6 Discussion

In this work, we rigorously test common intuitions about the benefits of deep ensembles to UQ and robustness, and find these explanations wanting. Below, we lay out limitations of our study, summarize our conclusions, and indicate important lines of future work.

**Ensembling in the overparametrized regime.** We emphasize that our analysis only focuses on ensembles of neural networks, and does not necessarily apply to ensembling techniques in general (e.g. random forests or gradient boosted decision trees). Indeed, we predict that many of our results are direct consequences of the fact that we are ensembling high-capacity "interpolating" models, which seem to generalize well despite being massively overparametrized [5, 1, 59, 29]. In future work, we will examine the effect of overparametrization directly by replicating these experiments with ensembles of weak learners.

**Neural network uncertainty quantification.** In examining the conditional distributions in Fig. 1, we see that OOD uncertainty quantification is not directly impacted by ensemble diversity. These findings show that the role of ensemble diversity in deep ensemble UQ is far more limited than previously hypothesized [e.g. 45, 18, 27].

**Effective robustness.** Our results in Figure 4 show that ensemble diversity does not yield improvements to robustness that cannot be explained by InD performance. This finding is in line with other results demonstrating that effective robustness is very difficult to achieve [2].

**When should we use deep ensembles?** Despite our results, we maintain that ensembling can be viewed as a reliable "black box" method of improving neural network performance, both InD and OOD. It is simple (though potentially expensive) to improve upon a model through ensembling, and training a single model that matches the performance of an ensemble is not always straightforward [40, 48, 74]. However we caution that deep ensembles are not a panacea for the issues faced by single models. In particular, it is dangerous to assume that deep ensembles mitigate the robustness issues of single models in contexts where we can expect dataset shift, or that ensemble diversity provides a reliable baseline for model uncertainty in the absence of ground truth. Thus, for many practitioners, the choice of using a deep ensemble versus a performance matched single model may ultimately be dictated by practical considerations, such as performance given a pre-determined parameter/FLOP budget for model training and evaluation [40, 48, 74]. Beyond these practical concerns, we have yet to find evidence for any reason to prefer the use of deep ensembles over an appropriately chosen single model.

## Acknowledgments and Disclosure of Funding

We thank John Miller for sharing models trained on CIFAR10, and Taori et al. [72] for making their trained ImageNet models and code open sourced and easy to use. We would also like to thank Dustin Tran for his insightful comments, and Julien Boussard for helpful discussions on statistical testing. TA is supported by NIH training grant 2T32NS064929-11. EKB is supported by NIH 5T32NS064929-13, NSF 1707398, and Gatsby Charitable Foundation GAT3708. GP and JPC are supported by the Simons Foundation, McKnight Foundation, Grossman Center for the Statistics of Mind, and Gatsby Charitable Trust.

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
