## A Societal impact

Deep ensembles are popular in many real world applications, and a potential negative impact of our work is to expose flaws in applications reliant upon deep ensembles, especially in adversarial settings like fraud detection (although this may lead to improved systems further on as well).

## B Code, data and compute

### B.1 Code and data

We provide general directions to reproduce the main results of our paper in the linked directions here: https://github.com/cellistigs/interp_ensembles#readme.

These directions reference two repositories, corresponding to two separate branches of our code-base. The "compare_performance" branch can be found here: https://github.com/cellistigs/interp_ensembles/tree/compare_performance. Likewise, the "imagenet_pl" branch can be found here: https://github.com/cellistigs/interp_ensembles/tree/imagenet_pl. This code repository, together with the instructions provided above, specify all training and visualization details relevant to our study.

Finally, we share relevant data as a Zenodo repository: https://zenodo.org/record/6582653#.Yo7R0y-B3fZ. This data provides the logit outputs from individual models (and some ensembles) on the in and out of distribution data that we consider. These data are referenced in the code above.

### B.2 Compute

We ran all CIFAR10 model training on Amazon Web Services (AWS), using the "p3.2xlarge" instance type with a Tesla V100 GPU. We ran half of ImageNet model training on an internal cluster with GeForce RTX 2080 Ti GPUs, and the other half on AWS with the "p3.8xlarge" instance type, again with Tesla V100 GPUs. Visualization and statistical testing was run on M1 MacBook Airs, and additionally on AWS "p3.2xlarge" and "p3.8xlarge" instances when additional capacity was required.

We show results for 50 models trained on CIFAR10, and 15 models trained on ImageNet. We estimate that on average, our CIFAR10 models required 3 hours of compute to train, and our ImageNet models required 48 hours to train. Finally, we estimate an additional 8 hours of compute required to run statistical tests and visualize results, resulting in a total of approximately 878 hours of total compute.

## C Decompositions for uncertainty metrics

### C.1 Jensen-Shannon divergence and entropic uncertainty

If we use Jensen-Shannon divergence (Eq. 2) as a metric for ensemble diversity [45, 18], we show ensemble uncertainty can be decomposed as:

$$\overbrace{\mathrm{H}\left[y \,|\, \bar{\boldsymbol{f}}(\boldsymbol{x})\right]}^{\text{ens. uncert.}} = \overbrace{\mathrm{JSD}_{p(\boldsymbol{f})}\left[y \,|\, \boldsymbol{f}(\boldsymbol{x})\right]}^{\text{ens. diversity}} + \overbrace{\mathbb{E}_{p(\boldsymbol{f})}\left[\mathrm{H}\left[y \,|\, \boldsymbol{f}(\boldsymbol{x})\right]\right]}^{\text{avg. single model uncert.}}. \tag{7}$$

where $\mathrm{H}[y \,|\, \cdot]$ represents the entropy of a categorical distribution parameterized by $(\cdot)$.

Furthermore, $\mathrm{JSD}_{p(\boldsymbol{f})}\left[y \,|\, \boldsymbol{f}(\boldsymbol{x})\right] = \frac{1}{M}\sum_{m=1}^{M} KL[y \,|\, \boldsymbol{f}(\boldsymbol{x})\|y \,|\, \bar{\boldsymbol{f}}(\boldsymbol{x})]$, the average KL divergence between individual model predictions and the ensemble prediction.

We write:

$$\mathrm{H}\left[y \mid \bar{\boldsymbol{f}}(\boldsymbol{x})\right] = -\frac{1}{C}\sum_i p(y_i \mid \bar{\boldsymbol{f}}) \log(p(y_i \mid \bar{\boldsymbol{f}}))$$

$$= -\frac{1}{C}\sum_i \frac{1}{M}\sum_j p(y_i \mid \boldsymbol{f}_j) \log(p(y_i \mid \bar{\boldsymbol{f}}))$$

$$= -\frac{1}{M}\sum_j \frac{1}{C}\sum_i p(y_i \mid \boldsymbol{f}_j) \log(p(y_i \mid \bar{\boldsymbol{f}}))$$

$$= -\frac{1}{M}\sum_j \frac{1}{C}\sum_i p(y_i \mid \boldsymbol{f}_j) \left[\log(p(y_i \mid \bar{\boldsymbol{f}})) - \log(p(y_i \mid \boldsymbol{f}_j)) + \log(p(y_i \mid \boldsymbol{f}_j))\right]$$

$$= -\frac{1}{M}\sum_j \frac{1}{C}\sum_i p(y_i \mid \boldsymbol{f}_j) \left[\log[\frac{p(y_i \mid \bar{\boldsymbol{f}})}{p(y_i \mid \boldsymbol{f}_j)}] + \log(p(y_i \mid \boldsymbol{f}_j))\right]$$

$$= -\frac{1}{M}\sum_j \frac{1}{C}\sum_i p(y_i \mid \boldsymbol{f}_j) \left[\log(\frac{p(y_i \mid \bar{\boldsymbol{f}})}{p(y_i \mid \boldsymbol{f}_j)})\right] + -\frac{1}{M}\sum_j \frac{1}{C}\sum_i p(y_i \mid \boldsymbol{f}_j) \log p(y_i \mid \boldsymbol{f}_j))$$

$$= \underset{p(\boldsymbol{f})}{\mathrm{JSD}}\left[y \mid \boldsymbol{f}(\boldsymbol{x})\right] + \underset{p(\boldsymbol{f})}{\mathbb{E}}\left[\mathrm{H}\left[y \mid \boldsymbol{f}(\boldsymbol{x})\right]\right]$$

### C.2 Variance and quadratic uncertainty

As in the main text, we provide a decomposition for a quadratic notion of uncertainty as:

$$U\left(\bar{\boldsymbol{f}}(\boldsymbol{x})\right) = \underset{p(\boldsymbol{f})}{\mathrm{Var}}\left[\boldsymbol{f}(\boldsymbol{x})\right] + \underset{p(\boldsymbol{f})}{\mathbb{E}}\left[U\left(\boldsymbol{f}(\boldsymbol{x})\right)\right] \tag{8}$$

where $U(\boldsymbol{f}(\boldsymbol{x}))$ is a quadratic notion of uncertainty:

$$U\left(\boldsymbol{f}(\boldsymbol{x})\right) \triangleq 1 - \sum_{i=1}^{C}\left[p(y = i \mid \boldsymbol{f}(\boldsymbol{x}))\right]^2.$$

And variance is defined as:

$$\underset{p(\boldsymbol{f})}{\mathrm{Var}}\left[\boldsymbol{f}(\boldsymbol{x})\right] = \sum_{i=1}^{C}\underset{p(\boldsymbol{f})}{\mathrm{Var}}\left[f^{(i)}(\boldsymbol{x})\right]$$

Then, the ensemble uncertainty can be decomposed as follows:

$$U\left(\bar{\boldsymbol{f}}(\boldsymbol{x})\right) = 1 - \sum_{i=1}^{C}\left[p(y = i \mid \bar{\boldsymbol{f}}(\boldsymbol{x}))\right]^2$$

$$= 1 - \underset{p(\boldsymbol{f})}{\mathbb{E}}\left[\sum_{i=1}^{C}\left[p(y = i \mid \boldsymbol{f}(\boldsymbol{x}))\right]\right]^2 + \underset{p(\boldsymbol{f})}{\mathbb{E}}\left[\sum_{i=1}^{C}\left[p(y = i \mid \boldsymbol{f}(\boldsymbol{x}))\right]\right]^2 - \sum_{i=1}^{C}\left[p(y = i \mid \bar{\boldsymbol{f}}(\boldsymbol{x}))\right]^2$$

$$= \underset{p(\boldsymbol{f})}{\mathbb{E}}\left[1 - \sum_{i=1}^{C}\left[p(y = i \mid \boldsymbol{f}(\boldsymbol{x}))\right]\right]^2 + \sum_{i=1}^{C}\underset{p(\boldsymbol{f})}{\mathbb{E}}\left[\left[p(y = i \mid \boldsymbol{f}(\boldsymbol{x}))\right]\right]^2 - \left[\underset{p(\boldsymbol{f})}{\mathbb{E}}\left[p(y = i \mid \boldsymbol{f}(\boldsymbol{x}))\right]\right]^2$$

$$= \underset{p(\boldsymbol{f})}{\mathbb{E}}\left[U\left(\boldsymbol{f}(\boldsymbol{x})\right)\right] + \underset{p(\boldsymbol{f})}{\mathrm{Var}}\left[\boldsymbol{f}(\boldsymbol{x})\right]$$

## D   Brier score Jensen gap

We consider the Brier Score of a single model:

$$\underset{\boldsymbol{f}}{\mathbb{E}}\left[B_p(\boldsymbol{f}_i)\right] = \underset{p(\boldsymbol{x},y)}{\mathbb{E}}\underset{\boldsymbol{f}}{\mathbb{E}}\left[\|\boldsymbol{f}_i(\boldsymbol{x}) - \mathbf{1}_y\|_2^2\right] \tag{9}$$

$$= \underset{p(\boldsymbol{x},y)}{\mathbb{E}}\left[\underset{\boldsymbol{f}}{\mathbb{E}}\left[\|\boldsymbol{f}_i(\boldsymbol{x})\|_2^2\right] + 2\bar{\boldsymbol{f}}(\boldsymbol{x})^\top \mathbf{1}_y + 1\right]$$

and the Brier score of the ensemble:

$$B_p(\bar{\boldsymbol{f}}) = \mathop{\mathbb{E}}_{p(\boldsymbol{x},y)} \left[ \|\bar{\boldsymbol{f}}(\boldsymbol{x}) - \mathbf{1}_y\|_2^2 \right] \tag{10}$$

$$= \mathop{\mathbb{E}}_{p(\boldsymbol{x},y)} \left[ \|\bar{\boldsymbol{f}}(\boldsymbol{x})\|_2^2 + 2\bar{\boldsymbol{f}}(\boldsymbol{x})^\top \mathbf{1}_y + 1 \right]$$

Note that Eq. (9) and Eq. (10) only differ by a single term:

$$B_p(\bar{\boldsymbol{f}}) = \mathop{\mathbb{E}}_{\boldsymbol{f}}[B_p(\boldsymbol{f})] + \mathop{\mathbb{E}}_{p(\boldsymbol{x})}\left[\|\bar{\boldsymbol{f}}(\boldsymbol{x})\|_2^2\right]$$

$$- \mathop{\mathbb{E}}_{p(\boldsymbol{x})} \mathop{\mathbb{E}}_{\boldsymbol{f}}\left[\|\boldsymbol{f}(\boldsymbol{x})\|_2^2\right]$$

$$= \mathop{\mathbb{E}}_{\boldsymbol{f}}[B_p(\boldsymbol{f})] - \mathop{\mathbb{E}}_{p(\boldsymbol{x})}[\|\mathop{\mathbb{E}}_{\boldsymbol{f}}\boldsymbol{f}(\boldsymbol{x})\|_2^2 - \|\bar{\boldsymbol{f}}(\boldsymbol{x})\|_2^2]$$

$$= \mathop{\mathbb{E}}_{\boldsymbol{f}}[B_p(\boldsymbol{f})] - \mathop{\mathbb{E}}_{p(\boldsymbol{x})}[\mathop{\mathbb{E}}_{\boldsymbol{f}}[\|\boldsymbol{f}(\boldsymbol{x})\|_2^2] - \|\mathop{\mathbb{E}}_{\boldsymbol{f}}[\boldsymbol{f}(\boldsymbol{x})]\|_2^2]$$

$$= \mathop{\mathbb{E}}_{\boldsymbol{f}}[B_p(\boldsymbol{f})] - \mathop{\mathbb{E}}_{p(\boldsymbol{x})}[\mathop{\mathrm{Var}}_{p(\boldsymbol{f})}[\boldsymbol{f}(\boldsymbol{x})]]$$

Where $\mathrm{Var}_{p(\boldsymbol{f})}[\boldsymbol{f}(\boldsymbol{x})]$ is:

$$\mathop{\mathrm{Var}}_{p(\boldsymbol{f})}[\boldsymbol{f}(\boldsymbol{x})] = \sum_{j=1}^{C} \mathop{\mathrm{Var}}_{p(\boldsymbol{f})}\left[f^{(j)}(\boldsymbol{x})\right]$$

.

We note that this relation holds at the level of individual data points as well.

## E   Expected behavior of Bayesian model average on InD/OOD uncertainty quantification

As a motivating example, we consider uncertainty quantification on InD and OOD data using a Bayesian model average, and relate our findings back to the implications presented in Sec. 4.5. An ideal Bayesian model average should express higher posterior variance on OOD data than InD data, even after controlling for other sources of uncertainty. To demonstrate this desired behavior in practice, we consider Gaussian processes, a class of models well regarded for its uncertainty quantification capabilities [65]. The Gaussian process model $f(\cdot)$ is defined by the following generative process:

$$p(f(\cdot)) = \mathcal{GP},$$
$$p(y \mid f(x)) = \mathcal{N}(0, \sigma^2(x)) \tag{11}$$

where $\sigma^2(x)$ is a heteroskedastic noise function defined as $\sigma^2(x) = \sin^2(x) + 0.01$. After conditioning on training data $\mathcal{D}$, the BMA at a test point $x$ is given by:

$$p(y \mid x, \mathcal{D}) = \mathcal{N}(\mu_{f|\mathcal{D}}(x), \mathrm{Var}_{f|\mathcal{D}}(x) + \sigma^2(x)), \tag{12}$$

where $\mu_{|\mathcal{D}}(\cdot)$ and $\mathrm{Var}_{f|\mathcal{D}}(\cdot)$ are the posterior predictive GP mean and variance, respectively, which can both be computed in closed form. (See [65] for closed-form expressions for these two functions). Crucially, the predictive variance in Eq. (12) is a uncertainty estimate that decomposes into epistemic and aleatoric components: the **posterior variance** term $(\mathrm{Var}_{f|\mathcal{D}}(\cdot))$ and the **likelihood variance** term $(\sigma^2(x))$, respectively.

In Fig. 5 (left), we generate a one-dimensional dataset by drawing 25 random data points over $x \in [0, 5]$ using the generative process defined in Eq. (11).[2] After fitting a GP model to these data, we compute the predictive posterior over the range $x \in [-5, 5]$. The points in $[0, 5]$ represent InD data—as they share the same domain as the training data—while the points in $[-5, 0]$ (orange) represent OOD data. In Fig. 5 (right), we observe that OOD predictions have much higher expected posterior variance, even after conditioning on a prediction's likelihood uncertainty. Note that this is in stark contrast to the analogous deep ensemble results in Sec. 4, where there is little to no conditional difference between OOD and InD predictions.

---

[2]In all experiments, the prior GP model has zero mean and a RBF covariance function with a lengthscale of 1.

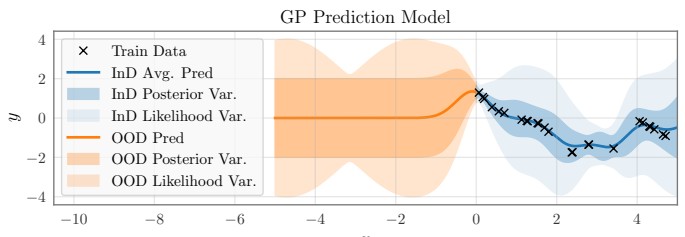
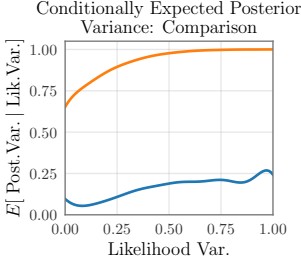

Figure 5: Example of a model where OOD predictions have higher posterior variance, even after controlling for other sources of uncertainty. **Left:** The predictive uncertainty expressed by a Gaussian process model on a toy regression dataset. OOD data (orange) express higher posterior variance than InD data (blue). **Right:** The expected posterior variance ) conditioned on a prediction's likelihood variance is also significantly larger for OOD data.

# F   Quantifying conditional diversity

In this section, we provide additional experimental details for the results in Fig. 1, and extend to other datasets and measures of ensemble diversity. We also introduce quantifications and signficance tests to validate the stability of our conclusions across many combinations of OOD dataset and model.

## F.1   Marginal distribution of average single model uncertainty

We end Sec. 4.2 with the surprising conclusion that any changes to ensemble UQ between InD and OOD data must come from changes in the distribution of average single model uncertainty, $p(\mathbb{E}(U(f(x)))$. Here we confirm empirically that this distribution does shift towards higher uncertainty on OOD data, for the same models that we present in Sec. 4.2. This shift drives any changes in ensemble diversity that we observe in practice.

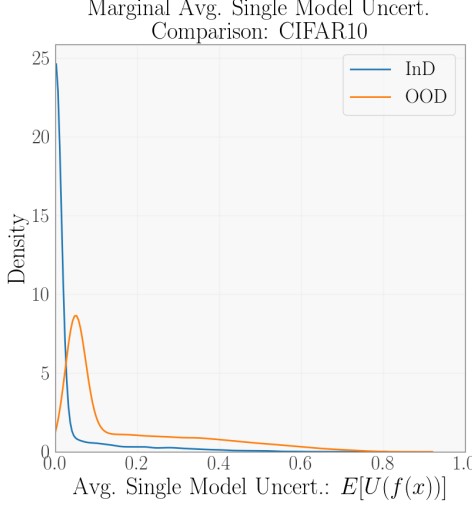
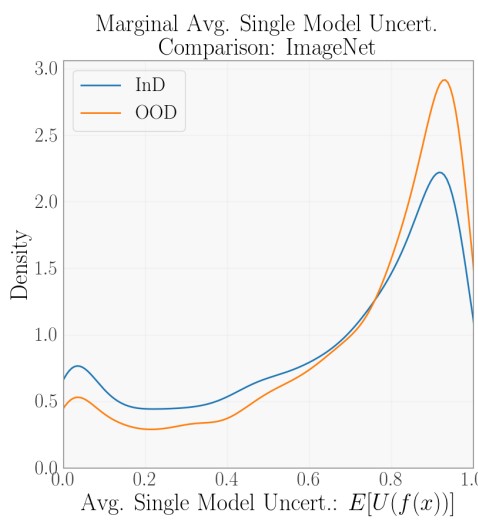

Figure 6: Distributions of average single model uncertainty for the WideResNet 28-10 ensembles trained on CIFAR10 (left) and the AlexNet ensembles (right), as in Fig. 1. InD and OOD test datasets are CIFAR10 and CINIC10 for the left panel, and ImageNet and ImageNet V2 for the right.

### F.2 Generating conditional distributions and conditional expectations

In order to depict conditional variance distributions, we fit kernel density estimates to the joint distribution of ensemble diversity and average single model uncertainty for all evaluation datasets. We generated KDEs with the bandwidth suggested by Scott's Rule, and approximate conditional distributions by dividing each column of our KDE grid by the average value.

To validate comparisons between conditional distributions more precisely, we estimate the conditional expectation $\mathbb{E}[\text{Diversity} \mid \text{Avg}]$ by fitting a Kernel Ridge Regression model to these data, giving the best fit curve to predict values of ensemble diversity from a given value of average single model uncertainty. We used a Gaussian kernel, with bandwidth identical to what was used to generate KDE plots.

Strictly to ease visualization, we generated conditional expectation estimates for CINIC10 with a randomly subsampled set of 10000 points when fitting Kernel Ridge Regression. We account for any potential bias this may introduce in our statistical quantifications below.

### F.3 Visualizations for other datasets and metrics

Figure 7 first shows the variance analysis that we conducted extended to CIFAR10/CIFAR10.1, estimated with an ensemble of 5 VGG 11 networks. In the rows below, we show all analogous conclusions for Jensen Shannon Divergence as a measure of ensemble diversity, instead of variance for the same models (ensembles of $M = 4$ VGG-11, WideResNet28-10, and AlexNet models for CIFAR10.1, CINIC10 and ImageNet V2 respectively). Across all datasets, we observe that the same trends hold as reported in Fig. 1. Namely, ensemble diversity is higher on OOD data than InD data, but that the corresponding conditional distributions are not distinguishable.

### F.4 Large scale quantification and statistical tests

In order to scale these analyses further, we devised a test statistic to directly compare the conditional expected diversity measures of InD and OOD data. Given conditional expectations for InD and OOD data, consider the following statistic:

$$d(InD, OOD) = \int d\text{Avg} \frac{\mathbb{E}_{OOD}[\text{Diversity} \mid \text{Avg}] - \mathbb{E}_{InD}[\text{Diversity} \mid \text{Avg}]}{\mathbb{E}_{InD}[\text{Diversity} \mid \text{Avg}]}$$

Intuitively, this statistic measures the percentage change in area under the conditional expectation curve when we consider an OOD conditional expectation instead of a corresponding InD conditional expectation.

We approximated this percentage increase in expected conditional diversity as sum of pointwise differences between InD and OOD, divided by the sum of the InD curve, and report results for all model and dataset pairs that we tested in Fig. 8,Fig. 9. Altogether, we see that in most cases, the percentage increases in area under the OOD curve are very small (for reference, the main text examples demonstrate changes on the order of $\sim 1\%$.) Although there are few sporadic cases where certain datasets demonstrate sizeable increases in our statistic on OOD data (consider variance for DenseNet 169 on CIFAR10-C Gaussian Noise, Severity Level 5), we note that these trends are inconsistent across individual models and datasets, limiting practical use of differences in OOD estimation. Furthermore, we note that our results on natural corruptions (leftmost two columns) are far more consistent than our results on synthetic corruptions (all others). In line with previous work [72], we prioritize results on natural corruptions in reporting our results.

Next, we performed Monte Carlo permutation tests to quantify the significance of the statistics that we observed:

- For each model and dataset upon which we computed a statistic, we first aggregated all datapoints from in and out of distribution model evaluations, and randomly permuted the order of these samples, generating a surrogate sample.
- We then refit Kernel Ridge Regression to the surrogate sample, and calculated the $d$ statistic that resulted.
- We calculated if the computed $d$ statistic was greater than or less than what we observed on our original sample.

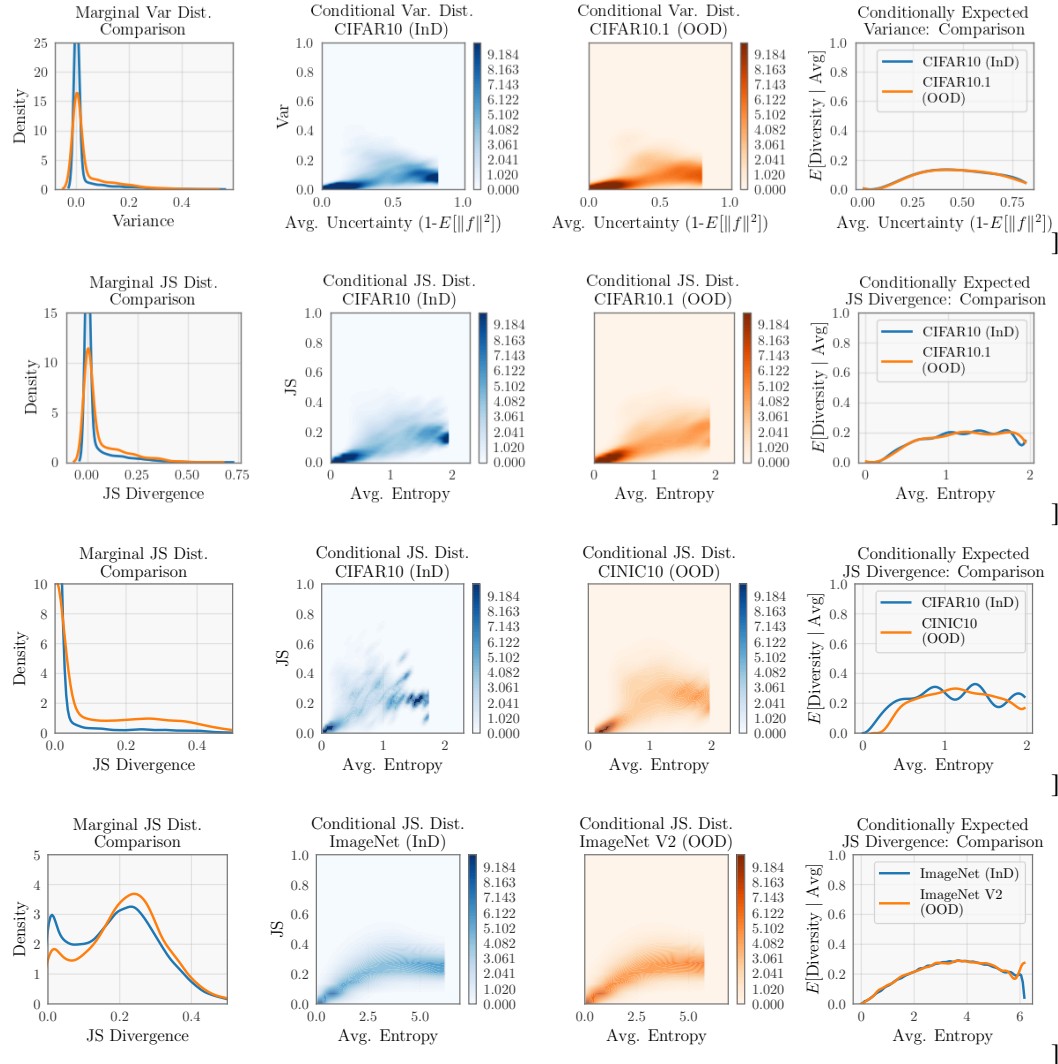

Figure 7: The top panels illustrates the InD vs OOD Variance for Cifar 10 vs Cifar10.1 with an ensemble of 4 VGG-11 networks. The bottom 3 panels illustrate the JS divergence on InD (Blue) and OOD (orange) data for CIFAR10 vs CIFAR10.1 (VGG-11), CIFAR10 vs CINIC10 (WideResNet-28-10) and ImageNet vs ImageNetV2 (AlexNet). Conventions and conclusions as in Figure 1.

- We repeated this process for a total of 100 surrogate samples.

From this process, we can treat the proportion of surrogate samples that exceeded the value of our true test statistic as a p value for the null hypothesis that the d statistic we calculated measures a significant difference between our two original samples (and in particular, that the conditional expectation of ensemble diversity on OOD data is significantly greater than that of ensemble diversity in InD data.)

In order to compute kernel ridge regression efficiently, we used GPytorch [23] with kernel partitioning to refit models many times on a GPU. This process allowed us to compute statistics on the entire CINIC10 evaluation set, alleviating all possibilities for error in visualization due to subsampling.

In Fig. 10 and Fig. 11, we report the estimated p values from this process. Our main goal is to communicate that in many cases, we found that the differences between conditional expectations for in and out of distribution data were almost certainly not significant, regardless of their absolute magnitude.

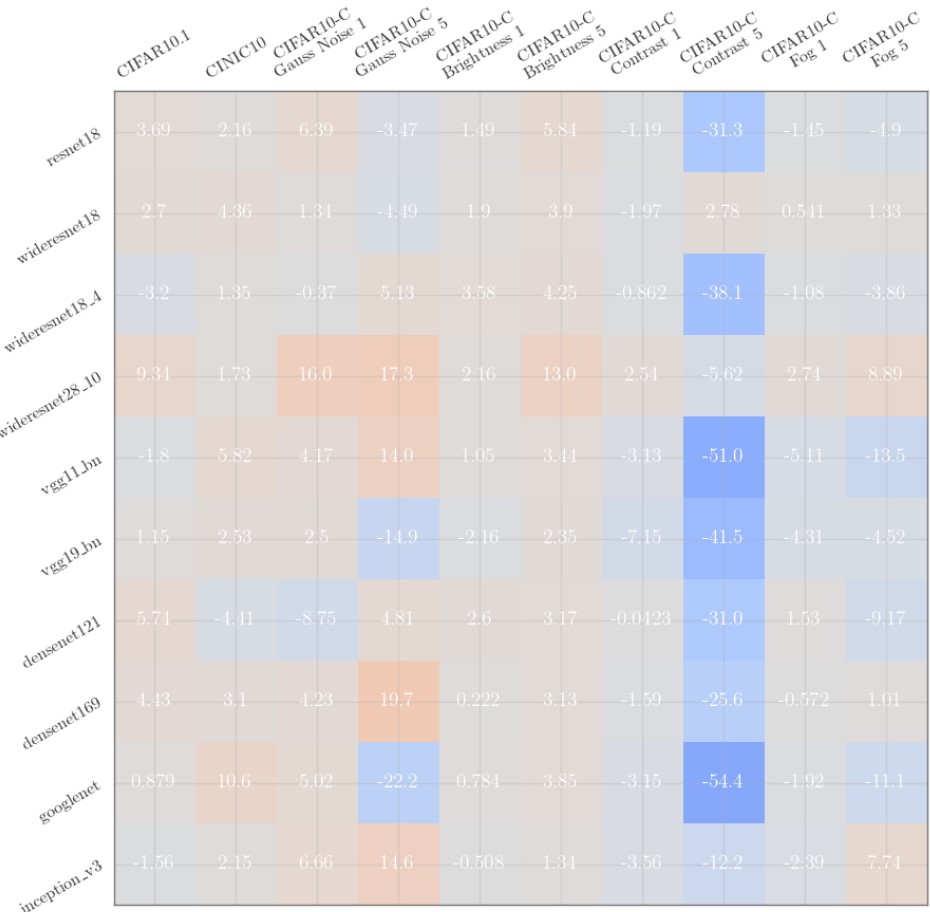

Figure 8: Percent Increase (OOD over InD) for Variance Decomposition

Finally, we show percentage increases for Imagenet on analogous $M = 5$ ensembles of AlexNet, ResNet 50, and ResNet 101 models Fig. 10, Fig. 11- on ImageNet V2, we once again fail to see any considerable increase on the conditional distributions of OOD data relative to InD data, regardless of metric.

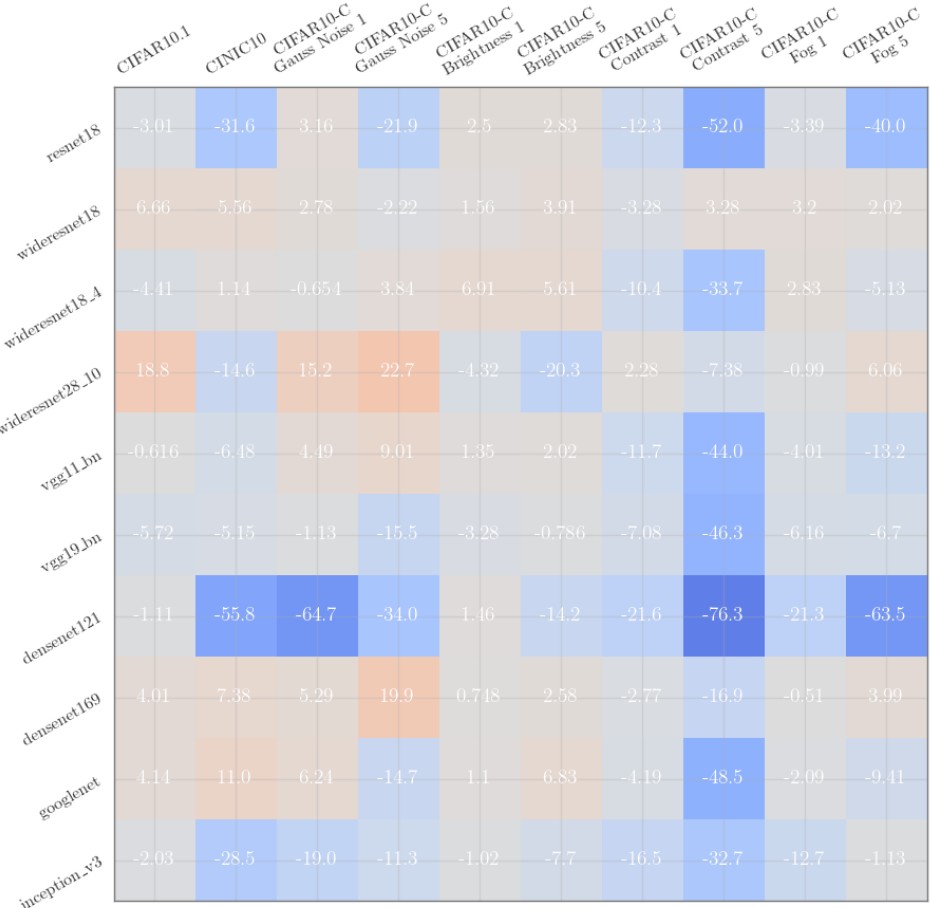

Figure 9: Percent Increase (OOD over InD) Jensen Shannon Decomposition

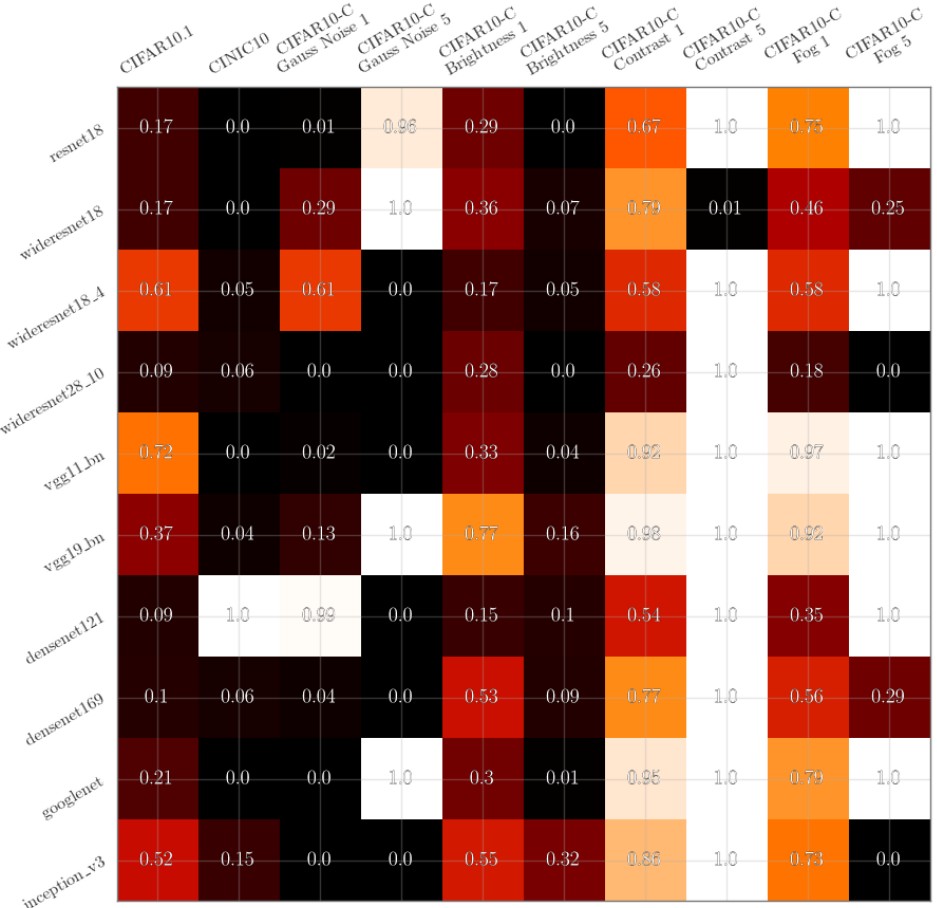

Figure 10: P values of (OOD over InD) difference for Variance Decomposition

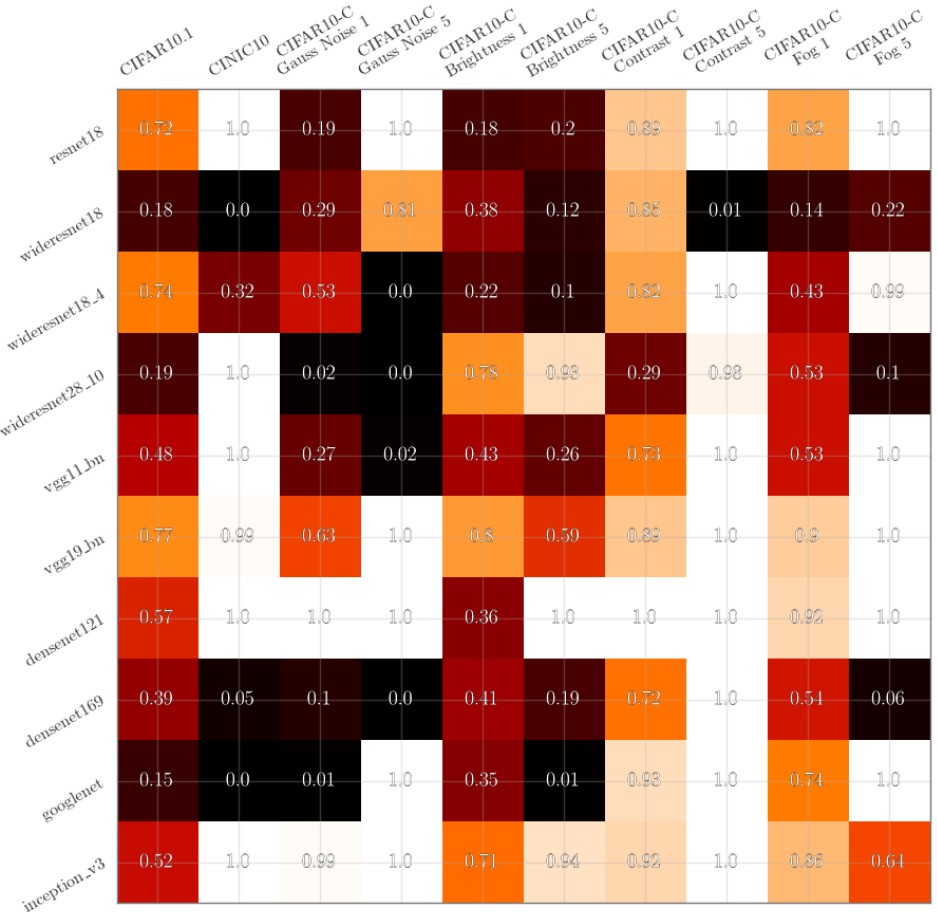

Figure 11: P values of (OOD over InD) difference for Jensen Shannon Decomposition

We can replicate the finding that differences between in and out of distribution test sets are quite small in the ImageNet dataset as well:

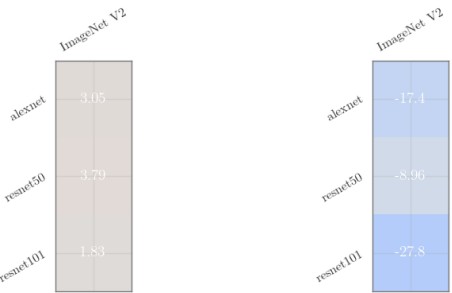

Figure 12: Percent Increase (OOD over InD) Variance Decomposition (left) and JS Divergence (right)

# G  Details of models for robustness experiments

We followed many of the same experimental procedures as [52] in order to generate ensembles for our experiments. We denote four main groups of models below:

## G.1  CIFAR10 models trained from scratch

We trained 10 different classes of models on CIFAR10, noted below. We used implementations from https://github.com/huyvnphan/PyTorch_CIFAR10 in order to train convolutional models adapted for CIFAR10 data sizes, with default hyperparameters, and manually extended existing implementations in this repo to create a WideResNet 18 with width 4.

- ResNet 18 [33]
- WideResNet 18-2, 18-4, 28-10 [78]
- GoogleNet, Inception v3 [71]
- VGG with 11 and 19 layers [69]
- DenseNet 121 and 169 [37]

We trained five independent instances of each of these architectures with random seeds for 100 epochs each (see code repo defaults for other hyper parameters.)

## G.2  CIFAR10 pretrained ensembles

We use the models trained by Miller et al. [52], and we thank the authors for graciously sharing these results with us.

## G.3  ImageNet models trained from scratch

We additionally trained two sets of ensembles from scratch on the ImageNet dataset. In particular, we trained 5 model ensembles of AlexNet and ResNet 101 models using implementations available at https://pytorch.org/vision/stable/models.html for 90 epochs each.

## G.4  Imagenet pretrained models

We use 5 of the ResNet50 models trained by [4] and the standard 78 trained models provided by Taori et al. [72].

# H  Additional generalization trend results

In this section, we report test statistics for the results we show in Fig. 4, and we extend the results from Fig. 4 to additional OOD datasets, namely CIFAR10.1 and ImageNet-C [34], illustrating generalization trends for ensembles and individual models for various distortions at different intensity levels. The results in this section show that for high intensity distortions, single models can break away from a well defined linear trend, as reported in [52]. However, even at the highest distortion levels, the generalization performance for ensembles and individual models heavily overlap, suggesting the lack of effective robustness demonstrated by deep ensembles is not dependent upon the same phenomena that generate strong trends in single models to begin with.

## H.1  Test statistics for generalization performance trends

In each table we report the regression coefficient (Coefficient), the standard error (Std. error) t-statistic, p-value and $R^2$ to reject the null hypothesis that there is no relation between InD and OOD performance for the different metrics considered (left column). The last column indicates the number of models (markers) for each model class depicted in Fig 4.

Note that we do not apply logit scaling to our axes as in [72], which was found to increase the fit of linear trend lines. Furthermore, we do not consider non-linear parametrizations of NLL, which could

potentially improve the quantification of overlap between single models and ensembles. We consider such parameterizations to be beyond the scope of this work.

Table 1: $R^2$ **for InD vs OOD generalization trend fits for different metrics**: CIFAR10 vs CINIC10 in Fig. 4a.

| Metric | Type | Coefficient | Std. error | t-statistic | p-value | R^2 | Number of models |
|---|---|---|---|---|---|---|---|
| 0-1 Error | All | 0.038 | 0.002 | 18.981 | 0.0 | 0.853 | 434 |
| | Single Model | 0.029 | 0.006 | 5.038 | 0.0 | 0.883 | 54 |
| | Ensemble | 0.039 | 0.002 | 18.349 | 0.0 | 0.848 | 380 |
| NLL | All | 0.116 | 0.006 | 18.285 | 0.0 | 0.894 | 434 |
| | Single Model | 0.120 | 0.022 | 5.511 | 0.0 | 0.864 | 54 |
| | Ensemble | 0.116 | 0.007 | 17.559 | 0.0 | 0.896 | 380 |
| Brier | All | 0.051 | 0.003 | 17.415 | 0.0 | 0.876 | 434 |
| | Single Model | 0.042 | 0.009 | 4.754 | 0.0 | 0.890 | 54 |
| | Ensemble | 0.052 | 0.003 | 16.754 | 0.0 | 0.873 | 380 |
| rESCE | All | 0.009 | 0.002 | 4.712 | 0.0 | 0.791 | 434 |
| | Single Model | 0.026 | 0.007 | 3.755 | 0.0 | 0.632 | 54 |
| | Ensemble | 0.007 | 0.002 | 3.860 | 0.0 | 0.801 | 380 |

Table 2: $R^2$ **for InD vs OOD generalization trend fits for different metrics**: ImageNet vs ImageNetV2 in Fig. 4b.

| Metric | Type | Coefficient | Std. error | t-statistic | p-value | R^2 | Number of models |
|---|---|---|---|---|---|---|---|
| 0-1 Error | All | 0.102 | 0.001 | 89.935 | 0.0 | 0.995 | 367 |
| | Single Model | 0.105 | 0.002 | 43.326 | 0.0 | 0.994 | 93 |
| | Ensemble | 0.101 | 0.001 | 78.643 | 0.0 | 0.995 | 274 |
| NLL | All | 0.432 | 0.008 | 54.749 | 0.0 | 0.989 | 367 |
| | Single Model | 0.443 | 0.018 | 24.091 | 0.0 | 0.984 | 93 |
| | Ensemble | 0.428 | 0.009 | 49.622 | 0.0 | 0.991 | 274 |
| Brier | All | 0.156 | 0.002 | 77.827 | 0.0 | 0.989 | 367 |
| | Single Model | 0.159 | 0.005 | 34.540 | 0.0 | 0.985 | 93 |
| | Ensemble | 0.156 | 0.002 | 69.984 | 0.0 | 0.991 | 274 |
| rESCE | All | 0.060 | 0.003 | 19.723 | 0.0 | 0.111 | 367 |
| | Single Model | 0.067 | 0.006 | 10.871 | 0.0 | 0.090 | 93 |
| | Ensemble | 0.058 | 0.004 | 16.342 | 0.0 | 0.113 | 274 |

Table 3: $R^2$ **for InD vs OOD generalization trend fits for different metrics**: CIFAR10 vs CIFAR10.1 in Fig. 13

| Metric | Type | Coefficient | Std. error | t-statistic | p-value | R^2 | Number of models |
|---|---|---|---|---|---|---|---|
| 0-1 Error | All | 0.038 | 0.002 | 18.981 | 0.0 | 0.853 | 434 |
| | Single Model | 0.029 | 0.006 | 5.038 | 0.0 | 0.883 | 54 |
| | Ensemble | 0.039 | 0.002 | 18.349 | 0.0 | 0.848 | 380 |
| NLL | All | 0.116 | 0.006 | 18.285 | 0.0 | 0.894 | 434 |
| | Single Model | 0.120 | 0.022 | 5.511 | 0.0 | 0.864 | 54 |
| | Ensemble | 0.116 | 0.007 | 17.559 | 0.0 | 0.896 | 380 |
| Brier | All | 0.051 | 0.003 | 17.415 | 0.0 | 0.876 | 434 |
| | Single Model | 0.042 | 0.009 | 4.754 | 0.0 | 0.890 | 54 |
| | Ensemble | 0.052 | 0.003 | 16.754 | 0.0 | 0.873 | 380 |
| rESCE | All | 0.009 | 0.002 | 4.712 | 0.0 | 0.791 | 434 |
| | Single Model | 0.026 | 0.007 | 3.755 | 0.0 | 0.632 | 54 |
| | Ensemble | 0.007 | 0.002 | 3.860 | 0.0 | 0.801 | 380 |

## H.2 Evaluation on other datasets

In this section we follow the same conventions as in Fig. 4 to analyze the generalization performance for two other OOD datasets for CIFAR10 and ImageNet, namely CIFAR10.1 and ImageNetC [34]. For ImageNetC we focus on our distortions from this dataset; namely brightness, contrast, fog and gaussian noise for three different degrees of corruption.

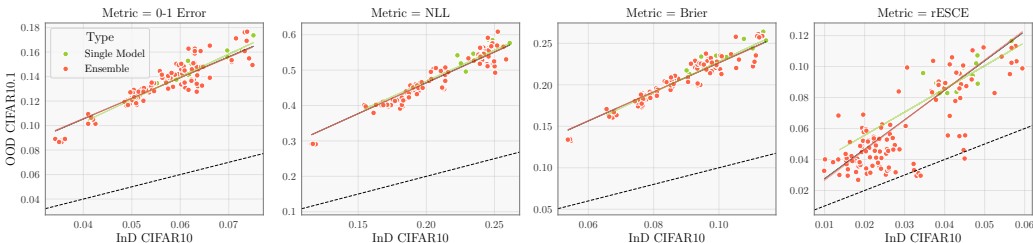

Figure 13: Generalization Trends for CIFAR10 vs CIFAR10.1.
Conventions and conclusions as in Fig. 4.

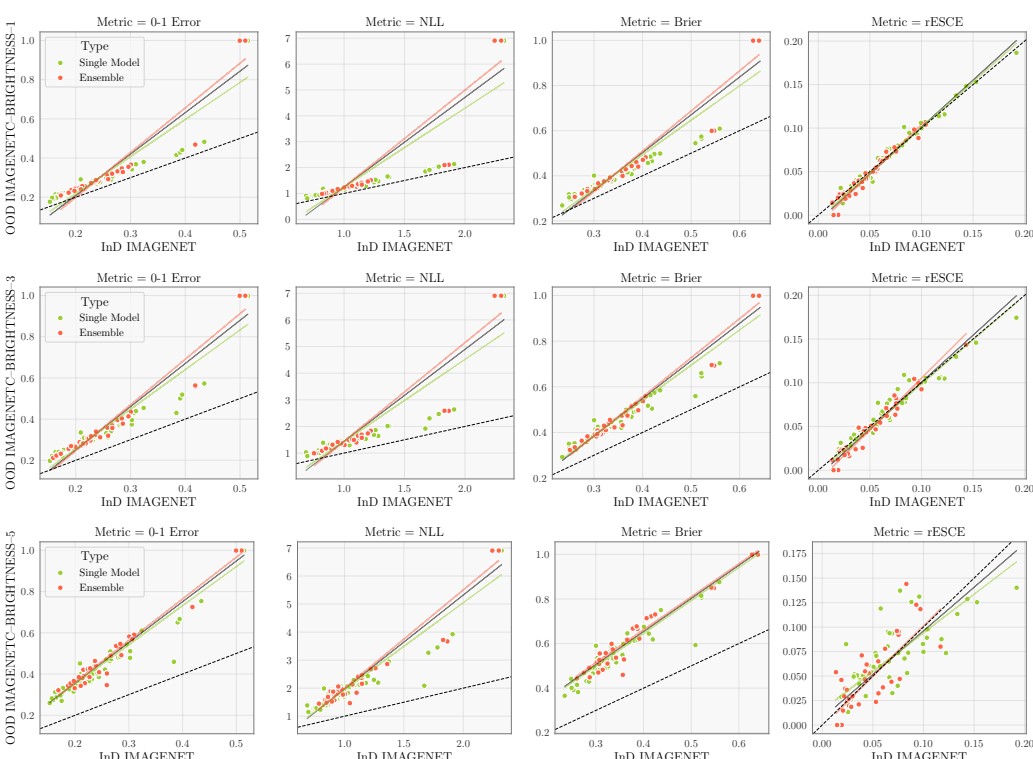

Figure 14: Generalization Trends for ImageNet vs ImageNet-C Brightness-1, 3 and 5.
Conventions and conclusions as in Fig. 4.

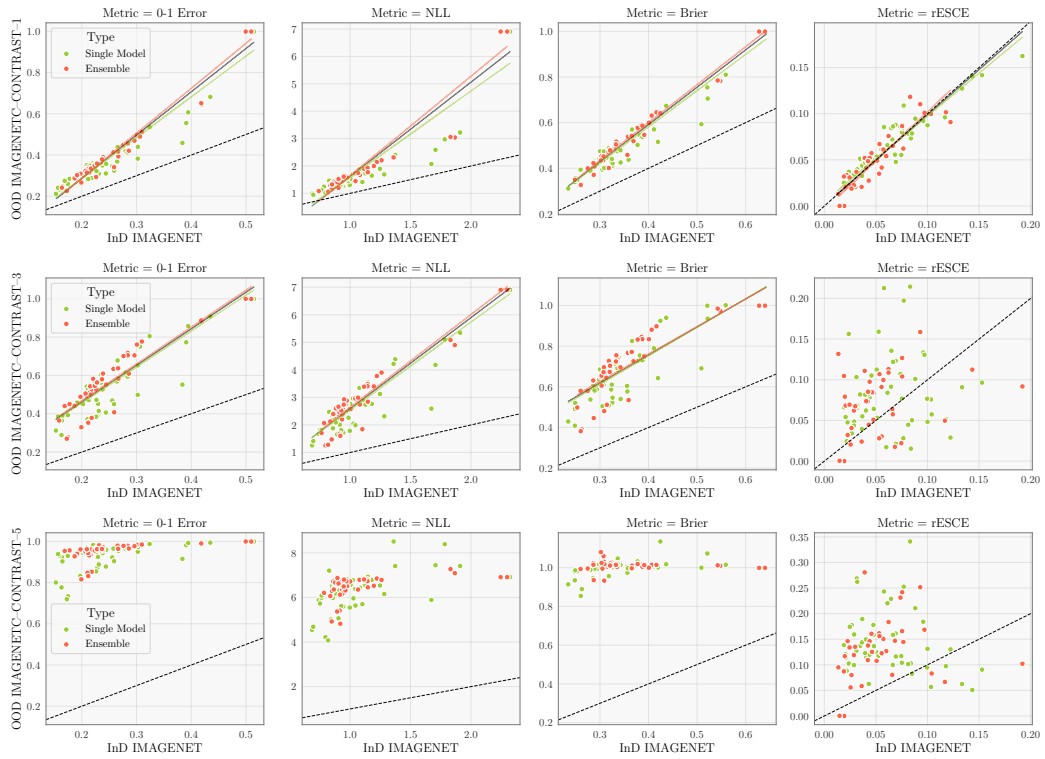

Figure 15: Generalization Trends for ImageNet vs ImageNet-C Contrast-1, 3 and 5.
Conventions and conclusions as in Fig. 4.

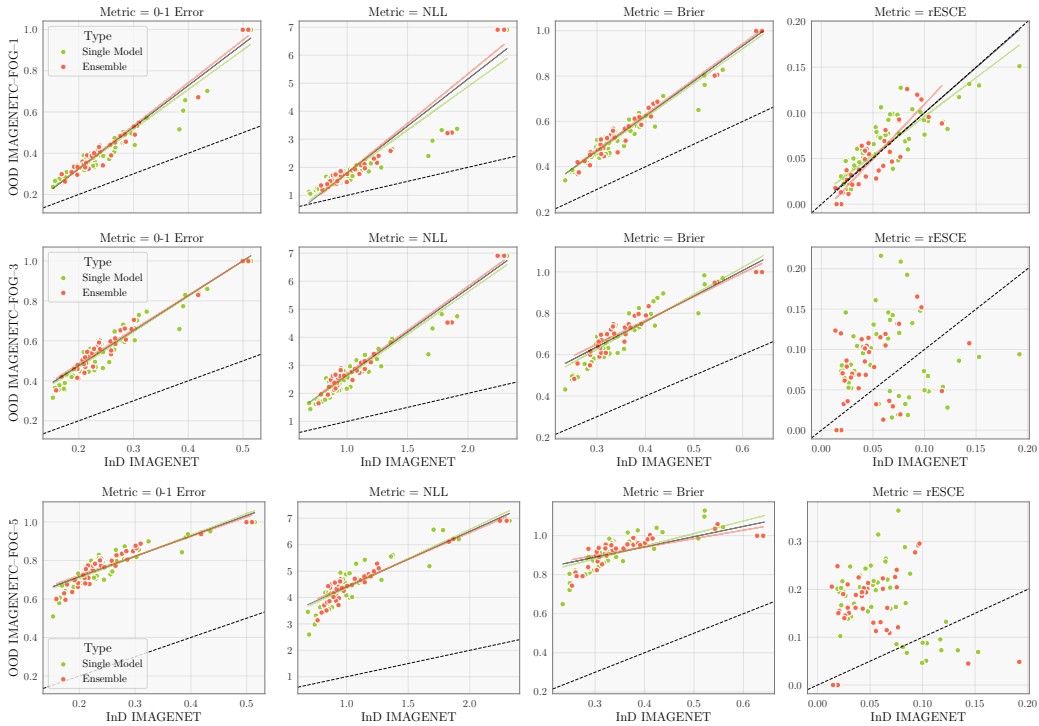

Figure 16: Generalization Trends for ImageNet vs ImageNet-C Fog-1,3, and 5.
Conventions and conclusions as in Fig. 4.

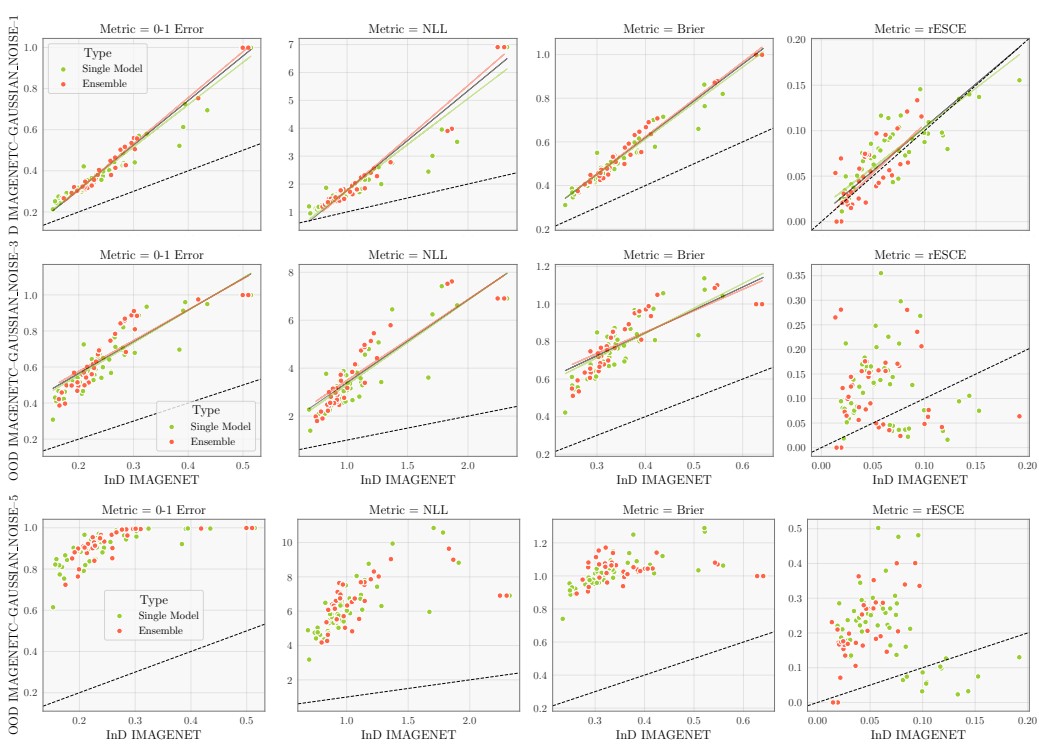

Figure 17: Generalization Trends for ImageNet vs ImageNet-C Gaussian Noise-1,3, and 5. Conventions and conclusions as in Fig. 4.

### H.3 Calibration metrics

The calibration error is a frequentist idea to measure the quality of uncertainty given by a model. The calibration error is given by,

$$\text{Calibration Error} = |Confidence - Accuracy|$$

To compare the calibration error across multiple models, practitioners have resorted to the expected calibration error (ECE) [58], which approximates the calibration error by binning the predictive probabilities and taking a weighted average of the calibration errors across bins. ECE provides a scalar summary statistic of the quality of uncertainty [26]. We employ both the ECE and, a smooth approximation, the root of the Expected Squared Calibration Error (rESCE) to compare the quality of uncertainty gained by ensembling over individual models. The rESCE is defined as,

$$\text{rESCE} = \sqrt{\sum \frac{\mid B_m \mid}{n}[acc(B_m) - conf(B_m)]^2}$$

Figure 18: Generalization trends for the Expected Calibration Error (ECE) for different datasets. Conventions and conclusions similar to the rESCE (right column) in Fig. 4.

Fig. 4 provides the rESCE for in distribution vs out of distribution for CIFAR10 vs CIFAR10.1 and Imagenet vs ImagenetV2. (See Appx. H.2 for additional datasets). In Fig. 18, we evaluate the generalization performance in terms of the ECE, following the conventions in Fig. 4. From Fig. 18 we see that there is no clear trend for InD versus OOD generalization across different datasets. Furthermore, we find that—for CIFAR10/CIFAR10.1—ensembles are able to achieve some amount of effective robustness with respect to the ECE metric. However, for the other two dataset pairs, we find that the ECE performance of ensembles heavily overlaps with the ECE performance of single models for most models.

### H.4 Comparison between homogeneous, heterogeneous and implicit deep ensembles

In this section we split the data from the ensemble model class in Fig. 4 into two sub classes: an ensemble class which now contains only the homogeneous ensembles, and the heterogeneous class; to explore if the ensemble model classes provide different generalization trends. We find that this is not the case for several in distribution and out of distribution pairs illustrated in Fig. 19.

We were additionally interested in evaluating whether implicit deep ensembles, models which aim to bridge the gap between individual networks and deep ensembles such as MC Dropout [21], Batch Ensemble [75], and MIMO [30], also follow the same observed trends. We include the performance of implicit ensembles, including MC Dropout and MIMO, in Fig. 19. The implicit deep ensemble models for ImageNet were constructed from a Resnet50 architecture, which was selected given its ubiquitous deployment, and availability in the open source Uncertainty Baselines library [57]. The number of implicit models considered for ImageNet is 12 (2 MC dropout models, and 10 for MIMO models). The implicit deep ensemble models for CIFAR10 were constructed from a WideResnet-28 architecture. In total, we considered 6 implicit ensemble models (3 MC dropout models and 3 MIMO models). Overall the results illustrated in Fig. 19 show that implicit deep ensembles also fall on the line, along with individual models, heterogeneous, and homogeneous ensembles, and do not constitute an effectively robust model class.

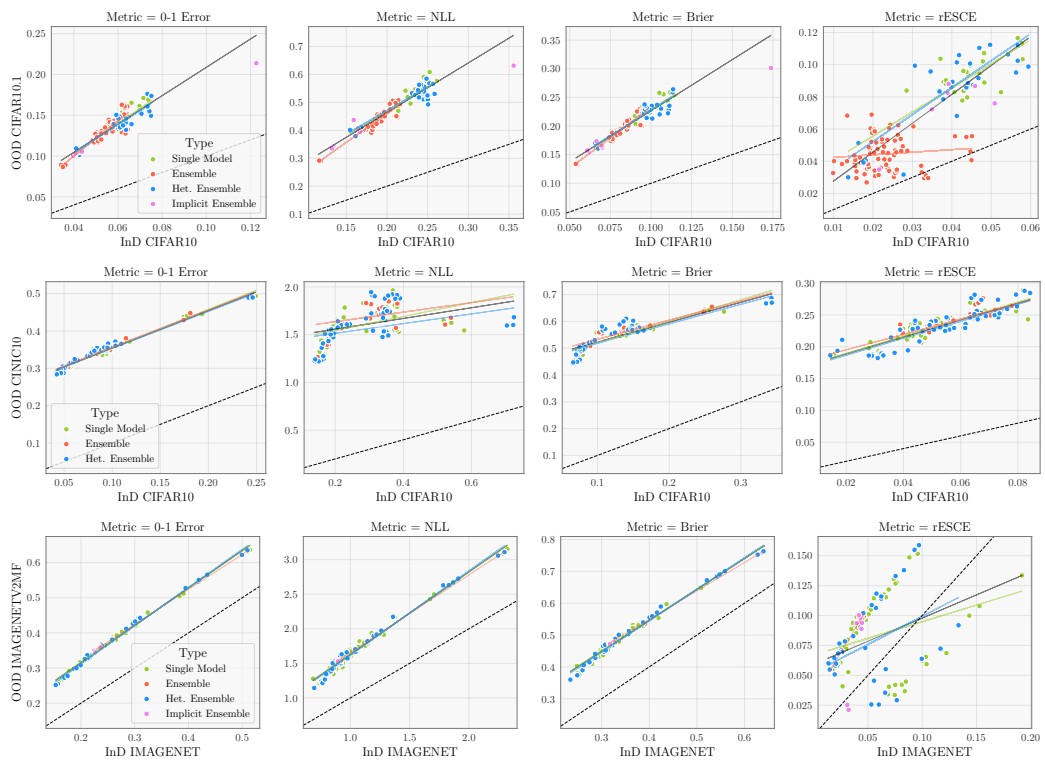

Figure 19: Homogeneous and heterogeneous deep ensembles in Fig. 4 follow similar generalization trends. Conventions and conclusions as in Fig. 4, including Implicit Deep ensemble models.

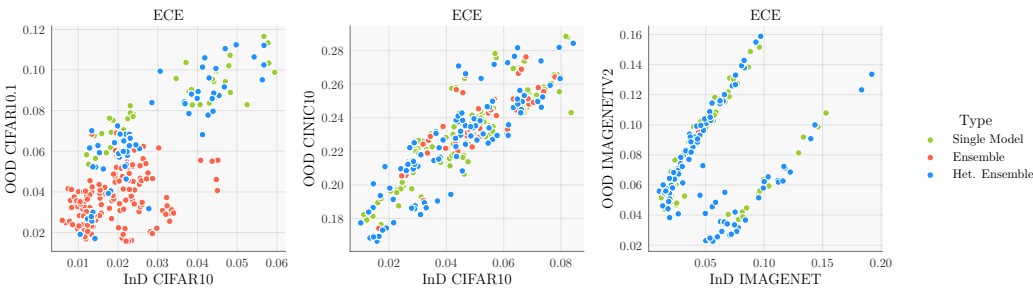

Figure 20: Expected Calibration Error (ECE) for homogeneous and heterogeneous ensembles follow similar trends as rESCE in Fig. 4.

# I Quantifying improvement similarity between single models and ensembles

In order to validate the statistical significance of the correlations that we observe between the improvements made by ensembles and single models, we considered as baselines the distribution of improvements we would expect from comparing *within* each model type- i.e. improvement correlations comparing the improvement of two performance matched single models, or two performance matched ensembles. Although Pearson's $R$ provides a good visual aid to interpret the trends visually, we wanted to be more agnostic when validating the trends that we see. We directly compared the improvement correlations that we see between ensemble/larger model pairs with improvement correlations that resulted when we substituted one member of these pairs with another, similarly performing model from the opposite model type (i.e., replace the ensemble with a control single model, or vice versa). We compared the resulting pair of improvement correlations with a kernel two sample test [25]. We calculated the unbiased test statistic $MMD_u^2$ from this paper for each combination of improvement correlations, and determined an appropriate threshold for these statistics based upon which we would reject the null hypothesis (Corollary 11 in [25]).

For each entry in the tables shown here, we consider the performance of four different kinds of predictions: an ensemble, an average single model with similar performance, a "control" set of average single models or ensembles (again with similar performance) and finally a base model, against which we are comparing the performance of all other models. Each entry compares one of the improvement correlations shown in Fig. 2 (given by the row) against the "control" improvement correlation listed in the column. In all comparisons, across InD and OOD data, CINIC10 and CIFAR10.1, and using NLL or Brier Score as metrics, we failed to reject the null hypothesis that the distributions we compared were significantly different. Each table shows the statistic value that we computed from any given pair of improvement correlations, along with the threshold statistic value we would have to exceed to reject the null hypothesis in parentheses. We list details of each comparison with each table.

Finally, at the end of this section we list the accuracies of the models that we compare, ensuring that the overall performance of these models does not differ too significantly, regardless of the metric under consideration.

Table 4: $MMD_u^2$ **to compare performance gains: CINIC10 NLL.** Base network: VGG-11. Average Model: WideResNet-18-4. Control Single Model: WideResNet-18-2. Control Ensemble: GoogleNet

|  |  | $\Delta$ Single/$\Delta$ Ctrl. Single | $\Delta$ Ensemble/$\Delta$ Ctrl. Ensemble |
|---|---|---|---|
| $\Delta$ Ensemble/$\Delta$ Single | CIFAR10 (InD) | 2.2e−3(0.069) | 2.1e−2(0.069) |
|  | CINIC10 (OOD) | 3.3e−3(0.031) | 2.0e−3(0.031) |

Table 5: $MMD_u^2$ **to compare performance gains: CINIC10 Brier Score**. Base network: VGG-11. Average Model: WideResNet-18-4. Control Single Model: WideResNet-18-2. Control Ensemble: GoogleNet

|  |  | $\Delta$ Single/$\Delta$ Ctrl. Single | $\Delta$ Ensemble/$\Delta$ Ctrl. Ensemble |
|---|---|---|---|
| $\Delta$ Ensemble/$\Delta$ Single | CIFAR10 (InD) | 2.7e−4(0.069) | 1.3e−2(0.069) |
|  | CINIC10 (OOD) | 4.2e−3(0.031) | 4.8e−3(0.031) |

Table 6: $MMD_u^2$ **to compare performance gains: CIFAR10.1 NLL**. Base network: ResNet 18. Average Model: WideResNet 18-4. Control Single Model: WideResNet 18. Control Ensemble: VGG 11

|  |  | $\Delta$ Single/$\Delta$ Ctrl. Single | $\Delta$ Ensemble/$\Delta$ Ctrl. Ensemble |
|---|---|---|---|
| $\Delta$ Ensemble/$\Delta$ Single | CIFAR10 (InD) | 2.4e−2(0.069) | 5.1e−3(0.069) |
|  | CIFAR10.1 (OOD) | 8.1e−3(0.15) | 2.0e−3(0.15) |

Table 7: $MMD_u^2$ **to compare performance gains: CIFAR10.1 Brier Score**. Base network: ResNet 18. Average Model: WideResNet 18-4. Control Single Model: WideResNet 18. Control Ensemble: VGG 11

| | | Δ Single/Δ Ctrl. Single | Δ Ensemble/Δ Ctrl. Ensemble |
|---|---|---|---|
| Δ Ensemble/Δ Single | CIFAR10 (InD) | 2.1e−2(0.069) | 2.5e−3(0.069) |
| | CIFAR10.1 (OOD) | 8.1e−3(0.15) | 8.7e−4(0.15) |

Table 8: **Corresponding accuracies for models compared on CIFAR10/CINIC10.** Architectures of models left to right: VGG-11, WideResNet-18-4, WideResNet-18-2, GoogleNet

| Dataset | Ensemble | Single Model | Single Model Control | Ensemble Control |
|---|---|---|---|---|
| CIFAR10 (InD) | 93.44% | $94.32 \pm 0.1495\%$ | $93.93 \pm 0.0804\%$ | 93.68% |
| CINIC10 (OOD) | 67.88% | $68.84 \pm 0.3936\%$ | $68.59 \pm 0.2762\%$ | 69.10% |

Table 9: **Corresponding accuracies for models compared on CIFAR10/CIFAR10.1.** Architectures of models left to right: ResNet18, WideResNet18-4, WideResNet18, VGG-11

| Dataset | Ensemble | Single Model | Single Model Control | Ensemble Control |
|---|---|---|---|---|
| CIFAR10 (InD) | 94.26% | $94.28 \pm 0.1430\%$ | $92.77 \pm 0.1839\%$ | 93.72% |
| CIFAR10.1 (OOD) | 86.10% | $86.26 \pm 0.1727\%$ | $84.61 \pm 0.4242\%$ | 84.12% |