# OpenReview forum: "Deep Ensembles Work, But Are They Necessary?"
_NeurIPS.cc/2022/Conference — NeurIPS 2022 Accept_

### Official Review · Reviewer_fE3e · 2022-07-07

**Rating:** 7
**Confidence:** 3
**Soundness:** 3 good
**Presentation:** 3 good
**Contribution:** 3 good

**Summary:**

The paper analyzes the causes of advantages of deep ensemble methods. Specifically, they show that:
1) Ensemble diversity does not contribute to an ensemble’s ability to detect OOD data. Improvements to UQ on OOD datasets are not due to changes in ensemble diversity.
2) OOD performance of deep ensembles are determined by their in-distribution (InD) performance and thus they do not provide an additional  "effective robustness".

**Questions:**

Can you explain what this "little benefit" is?
4.4."We conclude that, with regards to UQ and performance improvements, ensemble diversity offers little benefit over what can be obtained with single models."

**Ethics Review Area:**

["I don’t know"]

**Limitations:**

- It is positive that they point the following limitation of the paper out and include in the paper that they have not done such experiments:
"We emphasize that our analysis only focuses on ensembles of neural networks, and does not necessarily apply to ensembling techniques in general (e.g. as applied to random forests or gradient boosted decision trees)."

**Strengths And Weaknesses:**

Strengths:
- There is a strong contribution to understanding the deep ensembles. The paper shows that ensemble diversity is not responsible for the robustness and the robustness of the model with the OOD data is explainable by the performance increase in InD data. The Paper also shows improvements to UQ on OOD datasets are not due to changes in ensemble diversity.
- The paper is written well and it is easy to follow.

Weakness:

-"Discussion: When should we use deep ensembles?" Either this is not a very good title or the question itself is not well answered in the discussion. Something else is discussed under this title.

---

> ### Author Response · Authors · 2022-08-02
> **Author response**
>
> Thank you for your review and thoughtful recommendations.
>
> > "Discussion: When should we use deep ensembles?" Either this is not a very good title or the question itself is not well answered in the discussion. Something else is discussed under this title.
>
> Good point. We will include more guidelines along the following themes:
>
> - Practical considerations: even given identical performance, there could be benefits regarding parallelism, and efficiency of training an ensemble vs. a single neural network.
> - Estimating “equivalent models”: Previous work has identified situations where, given a fixed parameter budget, one can predict if an ensemble or a large single model will perform better on InD test data (Kondratyuk et al. 2020, Wasay and Idreos 2021, Lobacheva et al. 2021). We will suggest how these prior works, in conjunction with our results, might help practitioners choose models under a fixed parameter/FLOP budget.
>
> Nevertheless, we reiterate that our primary conclusion still holds: though ensembles can be more efficient in  terms of parallelism, parameter count, or number of FLOPs, they do not afford uncertainty quantification/robustness benefits that could not otherwise be achieved by single models.
>
> > Can you explain what this "little benefit" is? 4.4."We conclude that, with regards to UQ and performance improvements, ensemble diversity offers little benefit over what can be obtained with single models."
>
> We agree that the wording “little benefit” is confusing. We mean that this study found no significant benefit to deep ensembles that we could not obtain with single models. We will fix this in the revision.

---

> > ### Comment · Reviewer_fE3e · 2022-08-03
> > **Thanks for the reply.**
> >
> > Thank you for the reply.
> >
> > I would appreciate to see the promised text modifications in the camera-ready version.
> >
> > My concerns are adequately addressed.

---

> > > ### Author Response · Authors · 2022-08-05
> > > **Thank you!**
> > >
> > > Thank you very much for your response and your suggestions. We will make sure to include your suggested text modifications in our final version.

---

### Official Review · Reviewer_B3sc · 2022-07-08

**Rating:** 8
**Confidence:** 4
**Soundness:** 4 excellent
**Presentation:** 4 excellent
**Contribution:** 4 excellent

**Summary:**

The authors show with extensive empirical experiments that the power of ensemble models (in terms of effective OOD robustness and uncertainty quantification) can be replicated by a single larger model as well, therewith arguing the common belief that ensemble models are ways to achieve better performance as opposed to single models.

**Questions:**

Given the option to parallelize an ensemble of models during training, an ensemble might in some cases still be preferred, as also mentioned by the authors. Having a guideline in terms of how large a single model should be, would thus be very helpful. So can the authors give an indication how to know how large/complex a single model should be to match performance of an ensemble model? Did the authors now choose, for example, the same number of total trainable parameters for the single model as compared to the cumulative number of parameters of the ensemble?

Large models have the tendency to overfit on the training set, causing compromised test set performance. An ensemble of smaller models may have the advantage that the final result may be better, since either of the models that are part of the ensemble did not overfit that heavily. Is this a view the authors agree on? It might be an interesting addition to the discussion section of the paper.


**Limitations:**

Yes, this has been tackled.

**Strengths And Weaknesses:**

Originality:
This work does not propose a new idea, but challenges an existing belief. Which is original and brings a fresh view.

Quality:
The paper is of high quality. Many experiment, also in different settings, have been performed rigorously.

Clarity:
The paper is written clearly.

Significance:
It is important work that challenges a common belief about ensemble models. Even though, in practice, the use of ensembles might still remain popular, the learning points of this paper are useful for the community.


Minor point:
Eq. 2: I assume the superscript (i) refers to the class index in the C-dimensional vector. It was, however, not defined.

---

> ### Author Response · Authors · 2022-08-02
> **Author response**
>
> Thank you for the supportive remarks. We hope that the points we raise in the paper will invigorate further discussion around the usage of ensembles.
>
> > Minor point: Eq. 2: I assume the superscript (i) refers to the class index in the C-dimensional vector.
>
> You are correct that $y^{(i)}$ refers to class index $i$. We will clarify this notation.
>
> > Given the option to parallelize an ensemble of models during training, an ensemble might in some cases still be preferred... Having a guideline in terms of how large a single model should be, would thus be very helpful.
>
> This is a key question that we would like to investigate in future work. Previous work (Kondratyuk et al. 2020, Wasay and Idreos 2021, Lobacheva et al. 2021), suggests that - given a parameter or FLOP budget - ensembles of smaller models can achieve better InD accuracy than single larger models, but the results depend upon the task and parameter/FLOP budget. Looking forward, we aim to combine these prior works with our results on distribution shift to estimate how large a single model must be to match ensemble OOD performance.
>
> > Large models have the tendency to overfit on the training set, causing compromised test set performance. An ensemble of smaller models may have the advantage that the final result may be better, since either of the models that are part of the ensemble did not overfit that heavily.
>
> Given recent developments that suggest that large overparameterized models seem to generalize well (Belkin et al. 2019, Nakkiran et al. 2019, Adlam and Pennington 2020, Hastie et al. 2022), it is not immediately obvious whether ensembles of small models are less likely to overfit than single larger models. We will include a brief discussion on these conflicting intuitions in our revision. Nevertheless, our experiments compare ensembles versus (larger) single models with matching test set (in distribution) performance, and therefore the degree to which models overfit on test sets shouldn’t impact our results.

---

> > ### Comment · Reviewer_B3sc · 2022-08-09
> > **Response to authors**
> >
> > Thank you for the clarifications. I think that adding the above mentioned points to the discussion might be of interest to the reader and people that would like to build upon this research.
> > Further than that it's a good paper that should be accepted.

---

> > > ### Author Response · Authors · 2022-08-09
> > > **Thank you!**
> > >
> > > Thank you very much! We agree that the above points will be of interest to readers, and that addressing them in the discussion will improve our paper. We will be sure to include them in our final manuscript.

---

### Official Review · Reviewer_eLYu · 2022-07-09

**Rating:** 7
**Confidence:** 3
**Soundness:** 4 excellent
**Presentation:** 3 good
**Contribution:** 3 good

**Summary:**

This paper considers whether deep ensembles offer distinct benefits beyond individual models in uncertainty quantification (UQ) and out-of-distribution(OOD) detection. To study this problem, this paper empirically tests two hypotheses: 1)ensemble diversity is responsible for improved UQ; 2) ensemble diversity is responsible for improved (OOD detection) robustness. According to the empirical results, deep ensembles do not provide additional benefits to UQ and OOD detection as a single model with similar capacity.

**Questions:**

- In line 160, it is written that *ensemble uncertainty = ensemble diversity + average single model uncertainty* is a common metric. Can some references be provided?
- From my understanding, $\mathbb{f}=[f_1, \ldots, f_M]$. If this understanding is correct, can you provide the definition of $p(y=i \mid \mathbb{f(x)})$?
- In line 216, it is written that "the performance gains from ensembling are somehow fundamentally different than the performance gains from increasing a single model's capacity." It is not clear what that fundamental difference is.

**Limitations:**

This paper does not explicitly discuss its limitations.

**Strengths And Weaknesses:**

## Strength
- This paper proposes novel methods to verify whether ensemble diversity can bring additional benefits. Specifically, in testing the first hypothesis, this paper computes $E[var\mid E[U]]$ in terms of $E[U(f(x))]$. This demonstrates that the InD and OOD uncertainties estimated by deep ensembles are basically determined by the uncertainties of single models.
- This paper conducts comprehensive experiments, including homogeneous, heterogeneous, and implicit ensemble.
- This paper is significant in that it empirically demonstrates that previous commonly accepted understandings of deep ensemble benefits might not be totally correct.
- This paper explicitly discusses how its findings are connected to previous papers.

## Weakness
- Though this paper focuses on UQ and OOD detection, it is also good to report the accuracies of both ensemble methods and single models to ensure a fair comparison. For instance, in line 226, it will be good to report also the accuracies of ensemble and single model.

---

> ### Author Response · Authors · 2022-08-02
> **Author response**
>
> Thank you for your supportive remarks. We are unclear why your score reflects reservations in the paper when your review lists many strengths and few weaknesses. We address your questions below and hope that you consider increasing your score if you find our clarifications to be satisfactory.
>
> > Though this paper focuses on UQ and OOD detection, it is also good to report the accuracies of both ensemble methods and single models to ensure a fair comparison.
>
> This is a fair point. Though we match the ensemble/single models in Section 4.4 based on Brier score (our primary metric of concern, since we are examining the contribution of individual data points to the Brier score), our ensembles and single models also achieve remarkably similar accuracy. For example, the ensemble (ResNet 18) and single models (WideResNet-18-4) referenced on line 226 achieve the following accuracies:
>
> - InD: ensemble: 94.26% single models: 94.28+/-0.1430%
> - OOD: ensemble: 86.09% single models: 86.26+/-0.1727%
>
> We are happy to include an appendix table with the accuracies of all models compared in Section 4.4.
>
> > Ensemble uncertainty = ensemble diversity + average single model uncertainty... Can some references be provided?
>
> This uncertainty decomposition has been expressed for variance (Gustafsson et al. 2020, Kendall et al. 2017) as cited in the sentence immediately following the quoted text. The corresponding references for the analogous decomposition for Jensen Shannon divergence are provided in Appendix C.1 (Lakshminarayanan et al. 2017, Fort et al. 2019).
>
> > From my understanding, f=[f1,…,fM]. If this understanding is correct, can you provide the definition of p(y=i∣f(x))?
>
> We are unable to interpret the Latex formatting of this response in openreview, especially as regards to the symbol to the left of the first equality, which is also to the right of the conditioning bar in the provided conditional probability. If this symbol is $\boldsymbol{f}$, $\boldsymbol{f}$ represents a generic, single neural network, and $p(y=i|\boldsymbol{f})$ is just the softmax output probability of that network for class i. If this symbol is $\bar{f}$, this does indeed represent the ensemble, and $p(y=i\|\bar{f})$ is the ensemble prediction (given in eq 1) for class i. In any event, we will clarify this notation in the revision.
>
> > "The performance gains from ensembling are somehow fundamentally different than the performance gains from increasing a single model's capacity." It is not clear what that fundamental difference is.
>
> This statement describes a hypothesis that would have to be true conditional on the first part of that sentence: “If Var were also responsible for improving UQ…”. We refute it in section 4.3, providing evidence for the main claims of this paper (i.e. that benefits of ensembles are NOT somehow fundamentally different than benefits of choosing a larger single model). We will clarify this wording.

---

> > ### Comment · Reviewer_eLYu · 2022-08-05
> > **Thanks for the response**
> >
> > Thank you for the reply.
> >
> > For the formatting problem, my question is about $p(y = i | f(x)$ in line 163. From my understanding of the response, it refers to the softmax output from a single model.
> >
> > Except for this, all my questions are addressed. I will raise my score. It would also be great to see all mentioned modifications in the final version.

---

> > > ### Author Response · Authors · 2022-08-05
> > > **Thank you!**
> > >
> > > Thank you very much for your prompt response! Regarding notation, your interpretation is correct, and we will revise the relevant lines for clarity. We appreciate your suggestions throughout our manuscript, and we will make sure to address them all in the final version of the paper.

---

### Official Review · Reviewer_tPAQ · 2022-07-10

**Rating:** 7
**Confidence:** 5
**Soundness:** 3 good
**Presentation:** 3 good
**Contribution:** 2 fair

**Summary:**

This paper contributes an empirical analysis of deep ensembles. In particular, it seeks to understand whether an ensemble provides a distinct benefit over a single, larger model. The experiments on standard vision benchmarks cover both in-distribution and out-of-distribution performance.

**Questions:**

* In S4.4, how do you control overfitting / underfitting and hyperparameter tuning?
* In S5.2, it's not clear why ECE was not used. Can you offer a more detailed explanation?

**Limitations:**

I think the work would benefit from some more discussion of limitations, e.g. limited modalities considered.

**Strengths And Weaknesses:**

This paper makes a strong claim: "deep ensembles [...] do not provide benefits distinct from what could be achieved by a standard [larger] neural network." Section 4.4 provides some empirical evidence for this, but its comparing ensembles of ResNet18 to a single WideResNet-18-4 (a related but different architecture), so I'm not convinced this is sufficient. The other evidence is in the form of out-of-distribution detection / robustness experiments, where it's found ensembles have no consistent benefits over single models. However, it's already well-known that ensembles (like most neural networks) lack the necessary distance-awareness property necessary for principled uncertainty estimation (see eg https://arxiv.org/abs/2006.10108). In summary, (1) for iD experiments, this paper appears to be comparing apples and oranges, and it's not clear why the same architecture wasn't used (with different hyperparameters) to increase model capacity; and (2) for OOD, it's already known ensembles aren't to be trusted away from their training data. Therefore, I think this work would benefit from a reframing that emphases instead the paragraph on Line 68: "this paper does not disagree per se with prior claims about the benefits of ensembles," which is seemingly in conflict with the first quoted line above.

I didn't see much discussion of overfitting, which seems very relevant to the benefits (or lack thereof) of ensembles. Another common way to think about ensembles is as an approximate Monte Carlo estimate of the posterior over model parameters. I would have liked to see this interpretation of ensembles discussed, since it implies an orthogonal benefit over single models, namely accounting for the uncertainty of the network parameters. On the other hand, increasing the size of a single model would in general be expected to lead to increased overfitting, without additional regularization. It might be nice to discuss some Bayesian Ensembling work, such as "Approximately Bayesian Ensembling"

Overall, I think this work has many qualities (well-written, detailed experiments, considers an important question), but I think it suffers from two important downsides. First, I don't think the claims regarding ensembles in iD settings are supported by the experiments. Second, the remarks about deep ensembles in OOD settings (e.g., "We therefore suggest caution when using ensembles to differentiate sources of
uncertainty in downstream applications") are already well-established by prior work. Finally, the scope of the experiments is somewhat limited to a small number of vision datasets, and I would have liked to see the (quite broad) claims further supported by experiments on other modalities. In summary, it would be very interesting if the main claims of this paper are true, but to show that would require a significantly expanded S4.4.

---

> ### Author Response · Authors · 2022-08-02
> **Author response**
>
> We hope to address your concerns and clarify some potential misunderstandings about our paper.
>
> > Line 68: ("this paper does not disagree per se with prior claims about the benefits of ensembles") is seemingly in conflict with [the earlier claim that "deep ensembles... do not provide benefits distinct from what could be achieved by a standard [larger] neural network.]"
>
> We believe there is confusion with regard to these two claims, as they are not in conflict. Line 68 expresses the established insight (Lakshminarayanan et al. 2016, Ovadia et al. 2018, Fort et al. 2019, Wen et al. 2020) that - given a model architecture “A” - an ensemble of “A” models will offer better UQ/OOD robustness than a single “A” model. Conversely, our main claim (the second quote) is that _any_ single model (of a different architecture “B”) that matches the InD performance of the “A” ensemble will also achieve the same UQ and OOD robustness as the “A” ensemble. In other words, _deep ensembles offer no UQ/robustness advantages **after we control for their improved InD accuracy**_. We hope this explanation clarifies the relationship between these two claims, which we reiterate throughout the introduction, results, and discussion section.
>
> > Section 4.4 is comparing ensembles of ResNet18 to a single WideResNet-18-4 (a related but different architecture)… and it's not clear why the same architecture wasn't used (with different hyperparameters) to increase model capacity
>
> We believe this concern is due to the same confusion addressed in our previous comment. Because our main claim requires comparing deep ensembles of model architecture/training procedure (“A”) with single models of a different architecture/training procedure (“B”), experiments that compare a deep ensemble of ResNet18 to a single WideResNet-18-4 are direct tests of our main claim. It’s true that we could have instead matched ensemble/single model InD performance by changing hyperparameters, but this wouldn’t be any more appropriate than our current experiments.
>
> Additionally, please note that we include several other ensemble/single model comparisons in Section 4.4 and Appendix I.
>
> > It's already well-known that ensembles… lack the necessary distance-awareness property necessary for principled uncertainty estimation…
>
> While the distance awareness property is an important perspective, it is one part of a growing body of literature showing the limitations of deep ensemble UQ. For some other perspectives, see He et al. (2020), Osband et al. (2021), and Ciosek et al (2020). (We will discuss these prior works in the revision.) Despite these findings, deep ensembles remain a gold standard in high risk and safety critical settings (Gustafsson et al. 2020, Kim et al. 2021, Tran et al. 2022), and consequently our field needs _more_ perspectives that carefully scrutinize their UQ and robustness properties. Our work provides a novel perspective by comparing ensembles to larger single models, complimenting insights from prior works and further characterizing the limitations of ensemble UQ/robustness.
>
> > Another common way to think about ensembles is as an approximate Monte Carlo estimate of the posterior over model parameters. I would have liked to see this interpretation of ensembles discussed.
>
> We discuss the connection between ensembles and Bayesian methods in Appendix E, especially as it pertains to uncertainty quantification.
>
> > The scope of the experiments is somewhat limited to a small number of vision datasets, and I would have liked to see the (quite broad) claims further supported by experiments on other modalities…
>
> We disagree that our claims are too broad for our results. While we are interested in extending our results to other modalities, our findings hold on over 100 different model architectures/training modalities across 5 different dataset/shift pairs. This is comparable in scope with previous empirical studies on deep ensembles (Lakshminarayanan et al. 2016, Ovadia et al. 2018, Fort et al. 2019) and dataset shift (Recht et al. 2019, Taori et al. 2020, Miller et al., 2021).
>
> > How do you control overfitting / underfitting and hyperparameter tuning?
>
> All models are trained using default hyperparameter settings from their original papers or from pre-existing model testbeds, following the convention established by previous large scale studies (Taori et al., 2020, Miller et al. 2021). Importantly, all of our ensemble/single model comparisons control for InD test performance, so over/underfitting is not a relevant concern for our results.
>
> > In S5.2, it's not clear why ECE was not used.
>
> As we state in Appendix H.3, there is little correlation between a single model’s InD ECE and OOD ECE. Conversely, there is a strong correlation between a model’s InD/OOD rESCE, which establishes a more meaningful notion of “effective robustness” for calibration. Regardless, we show in Appendix H.3 that deep ensembles are not distinguishable from single models on ECE.

---

> > ### Comment · Reviewer_tPAQ · 2022-08-05
> > **Thanks for the response**
> >
> > The response has addressed some of my main concerns and my recommendation will be adjusted accordingly. My remaining concerns relate to how the work is positioned relative to previous results, such as the lack of 'distance awareness' of deep ensembles, which the authors say that they will address in a revised version of the paper. Thanks for the detailed response!

---

> > > ### Author Response · Authors · 2022-08-05
> > > **Thank you!**
> > >
> > > Thank you very much for your thoughtful response. We will make sure to revise our manuscript in accordance with your suggestions, and in our revision we will take particular care to position our work in the context of prior results, such as those that discuss distance awareness. These steps will meaningfully improve our paper.

---

### Author Response · Authors · 2022-08-02
**Thank you for your reviews!**

We thank the reviewers for their feedback regarding our manuscript. Overall, we are pleased that reviewers found this paper to be “comprehensive [in its] experiments” (eLYu), “important work that challenges a common belief” (B3sc), and “a strong contribution to understanding deep ensembles” (fE3e). We have responded individually to each reviewer in separate comments below.

---

### Meta-Review · Area_Chair_dZtW · 2022-08-31

**Recommendation:** Accept
**Confidence:** Certain

**Metareview:**

This paper challenges a widely held view in deep learning: that deep ensembles are always superior to single models when it comes to uncertainty quantification and robustness. The authors convincingly show that a single big model may do equally well, and that much of the benefit of ensembles over single models of the same size seems to derive more from their improved accuracy than from their diversity. The reviewers thoroughly investigated the authors' claims and engaged in discussion. All reviewers are convinced that the main claim of the paper is mostly correct, and that it is a very interesting finding. I therefore strongly recommend accepting this paper.

**Award:**

No

---

### Decision · Program_Chairs · 2022-09-14

Accept